# Functional heterogeneity within the rodent lateral orbitofrontal cortex dissociates outcome devaluation and reversal learning deficits

Marios C Panayi[1,2]*, Simon Killcross[1]

[1]School of Psychology, The University of New South Wales, Kensington, Australia; [2]Department of Experimental Psychology, University of Oxford, Oxford, United Kingdom

**Abstract** The orbitofrontal cortex (OFC) is critical for updating reward-directed behaviours flexibly when outcomes are devalued or when task contingencies are reversed. Failure to update behaviour in outcome devaluation and reversal learning procedures are considered canonical deficits following OFC lesions in non-human primates and rodents. We examined the generality of these findings in rodents using lesions of the rodent lateral OFC (LO) in instrumental action-outcome and Pavlovian cue-outcome devaluation procedures. LO lesions disrupted outcome devaluation in Pavlovian but not instrumental procedures. Furthermore, although both anterior and posterior LO lesions disrupted Pavlovian outcome devaluation, only posterior LO lesions were found to disrupt reversal learning. Posterior but not anterior LO lesions were also found to disrupt the attribution of motivational value to Pavlovian cues in sign-tracking. These novel dissociable task- and subregion-specific effects suggest a way to reconcile contradictory findings between rodent and non-human primate OFC research.

DOI: https://doi.org/10.7554/eLife.37357.001

*For correspondence:
marios.panagi@psy.ox.ac.uk

**Competing interests:** The authors declare that no competing interests exist.

## Introduction

The orbitofrontal cortex (OFC) in rodents and primates is critical for updating behaviour flexibly when outcome contingencies change (*Murray et al., 2007*). Compelling evidence for this view comes from studies using outcome devaluation procedures in which the value of a reward is reduced to test whether behaviour is updated to reflect changes in the outcome's current value. In rodents, OFC lesions disrupt the appropriate reduction in anticipatory responding for a reward that has been paired with illness and has become aversive (*Gallagher et al., 1999*; *Pickens et al., 2003*, *2005*). Similar conclusions have been reached in human fMRI studies (*Gottfried et al., 2003*) and non-human primate studies, where OFC function is disrupted by excitotoxic lesions and functional inactivation (*Izquierdo and Murray, 2004*, *2010*; *Izquierdo et al., 2004*, *2005*; *Machado and Bach-evalier, 2007*; *Rolls, 2000*; *West et al., 2011*).

Similar to outcome devaluation, reversal learning procedures involve updating behaviour when rewarded and non-rewarded task contingencies change. Although OFC lesions do not impact initial acquisition of rewarded and non-rewarded cues or actions, they significantly disrupt the flexible updating of behaviour following the reversal of these contingencies (*Boulougouris et al., 2007*; *Murray et al., 2007*; *Schoenbaum et al., 2003*). Both outcome devaluation and reversal learning require flexibly tracking changes in learned contingencies and updating behaviour appropriately when contingencies or outcome values change, and both procedures are disrupted by damage to the OFC.

A key requirement of any theory of OFC function is to account for the deficits in both outcome devaluation and reversal learning following disruption of OFC function (*Delamater, 2007*; *Murray et al., 2007*; *Rudebeck and Murray, 2014*; *Wikenheiser et al., 2017*; *Wilson et al., 2014*). However, the generality of these deficits has been questioned. For example, OFC lesions do not disrupt outcome devaluation in instrumental action-outcome learning procedures (*Balleine et al., 2011*; *Ostlund and Balleine, 2007a*), which is in contrast to robust effects in Pavlovian outcome devaluation (*Schoenbaum et al., 1999*). Furthermore, reversal deficits are not always reported following OFC lesions, and the nature of the deficit is not always consistent (*Boulougouris et al., 2007*; *Burke et al., 2009*; *Chudasama and Robbins, 2003*; *Mariano et al., 2009*; *McAlonan and Brown, 2003*; *Rudebeck and Murray, 2011b*; *Rudebeck et al., 2013*; *Schoenbaum et al., 2003*). Although it is tempting to attribute these differences to different task parameters between studies, we propose that these discrepancies are also likely to be caused by functional heterogeneity within regions classified as OFC.

Indeed, there is mounting evidence that the OFC is a functionally heterogeneous structure (*Izquierdo, 2017*; *Mar et al., 2011*; *Rudebeck and Murray, 2011a*). Notably, *Ostlund and Balleine, (2007a)* used focal lesions of the lateral OFC (LO), which did not disrupt instrumental outcome devaluation, whereas earlier Pavlovian devaluation studies in rodents involved widespread damage to the ventral (VO), lateral (LO), dorsolateral (DLO), anterior agranular insular (AI), and even medial (MO) OFC subregions (*Gallagher et al., 1999*; *Pickens et al., 2003*, *2005*). It is therefore unclear which subregions contribute to flexible behavioural control tested by Pavlovian outcome devaluation or reversal learning in rodents.

Here, we investigate whether the role of OFC in flexible behavioural control is specific to cue-guided (Pavlovian) and not action-guided (instrumental) behaviours (*Ostlund and Balleine, 2007a*) using focal LO lesions. Then we examine whether there is functional heterogeneity within the anterior-posterior plane of the LO region using Pavlovian outcome devaluation and reversal learning procedures.

## Results

### Instrumental devaluation by taste aversion

First, we tested whether the OFC plays a necessary role in guiding flexible action-outcome behaviour in an instrumental outcome devaluation task. In contrast to Pavlovian devaluation using taste aversion (*Gallagher et al., 1999*; *Pickens et al., 2003*, *2005*), *Ostlund and Balleine, (2007a)* have shown that OFC lesions do not disrupt behaviour in an instrumental devaluation task using specific satiety as the method of devaluation. We extend these findings to instrumental devaluation using taste-aversion as the method of devaluation.

Following recovery from sham or excitotoxic OFC lesions (*Figure 1A*, N = 32; sham devalued *n* = 8, sham non-devalued *n* = 8, lesion devalued *n* = 8, lesion non-devalued *n* = 8), half the animals in each lesion group were assigned to have an instrumental reinforcer devalued (devalued group) or an alternative reinforcer devalued (non-devalued group). Rats were trained to lever press for either pellet or liquid sucrose rewards on a random interval 30 s schedule (RI30), and were exposed to the alternative reward non-contingently in a separate session on each day of training. OFC lesions did not affect the lever pressing acquisition across the 3 days of RI30 acquisition training (*Figure 1B*). A mixed Lesion (sham, lesion) x Devaluation (devalued, non-devalued) x Day (3 days) ANOVA revealed only a significant main effect of Day ($F_{(2, 56)}$=13.99, p<0.001, all remaining $F < 1.31$, p>0.26). A significant linear trend of Day ($F_{(1, 28)}$=21.80, p<0.001) suggested that all groups increased lever responding across acquisition days.

Next, animals in the devalued groups acquired a taste aversion to the instrumental reinforcer following pairings with LiCl injections (in contrast to pairings of the alternate reinforcer with control saline injections) whereas animals in the non-devalued groups acquired a taste aversion to the alternative reinforcer and the instrumental outcome was paired with saline injections. Taste aversion was successfully acquired to the food paired with LiCl, as shown by decreased consumption compared to the food paired with saline injections (*Figure 1C*), and there were no apparent group differences in acquiring this taste aversion. A mixed Lesion x Devaluation (devalued, non-devalued group) x Pairing (3 pairings) x Injection (LiCl, saline) ANOVA supported the acquisition of taste aversion with

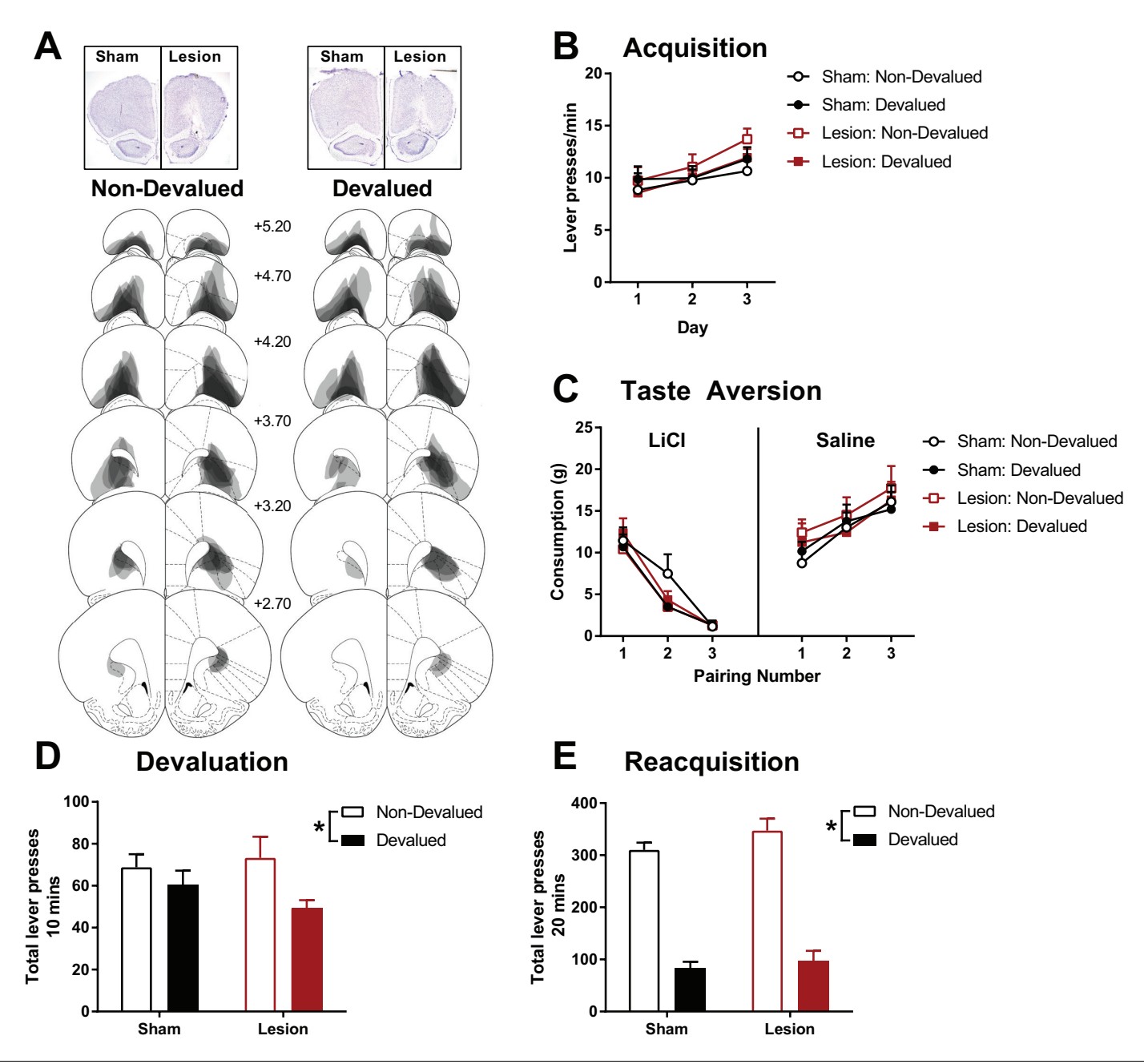

**Figure 1.** The effects of excitotoxic OFC lesions on instrumental devaluation by taste aversion. (**A**) Representative OFC lesion damage in the Non-Devalued (left) and Devalued (right) lesion groups. Semi-transparent grey patches represent lesion damage in a single subject, and darker areas represent overlapping damage across multiple subjects. Coronal sections are identified in mm relative to bregma (*Paxinos and Watson, 1997*). (**B**) Rate of lever pressing during 3 days of instrumental acquisition. (**C**) Mean reward consumption during taste aversion learning, consumption of rewards paired with LiCl induced nausea decreased across injection pairings (Left), whereas consumption of rewards paired with saline injections increased across injection pairings. (**D**) Total lever presses during the 10 mins devaluation test in extinction. Within-session responding presented in *Figure 1— figure supplement 1*. (**E**) Total lever presses during the 20 mins re-acquisition test with rewards delivered instrumentally. Error bars depict + SEM. (*) Symbol denotes statistical significance of simple or main effects following a significant interaction.
DOI: https://doi.org/10.7554/eLife.37357.002

The following figure supplement is available for figure 1:

**Figure supplement 1.** The effects of excitotoxic OFC lesions on instrumental devaluation by taste aversion.
DOI: https://doi.org/10.7554/eLife.37357.003

significant main effects of Pairing ($F_{(2, 56)}$=10.55, p<0.001), Injection ($F_{(1, 28)}$=35.96, p<0.001), and a Pairing x Injection interaction ($F_{(2, 56)}$=176.17, p<0.001, all remaining $F$ < 2.48, p>0.09). Similarly, a significant Pairing x Injection linear trend effect ($F_{(1, 28)}$=513.68, p<0.001) suggested that consumption of food paired with LiCl significantly decreased across pairings (linear trend across Pairing for LiCl, $F_{(1, 28)}$=286.92, p<0.001) whereas consumption of the food paired with saline increased (linear trend across Pairing for saline, $F_{(1, 28)}$=67.57, p<0.001). These findings support previous reports that OFC lesions do not affect initial learning of instrumental lever pressing behaviour, or sensitivity to acquiring taste aversions.

Devaluation of the instrumental response was then assessed by an extinction test of lever pressing. At test the groups with the devalued instrumental reinforcer performed fewer lever presses than the non-devalued groups (i.e. the groups with the alternative reinforcer devalued), but this was not differentially affected by lesion or sham surgery (*Figure 1D*). A univariate Lesion x Devaluation ANOVA revealed a significant main effect of Devaluation ($F_{(1, 28)}$=5.14, p=0.03) that did not significantly interact with Lesion (Lesion x Devaluation $F_{(1, 28)}$=1.19, p=0.29, main effect of Lesion $F_{(1, 28)}$=0.21, p=0.65). Therefore, a significant devaluation effect was found across both lesion groups (additional analysis of devaluation performance within the session presented in *Figure 1—figure supplement 1*).

Similar to the devaluation test in extinction, subsequent re-acquisition of the instrumental response with the delivery of the outcome was significantly affected by the acquired taste aversion showing a strong devaluation effect (*Figure 1E*). A univariate Lesion x Devaluation ANOVA revealed a significant main effect of Devaluation ($F_{(1, 28)}$=181.01, p<0.001) that did not significantly interact with Lesion (Lesion x Devaluation $F_{(1, 28)}$=0.43, p=0.52, main effect of Lesion $F_{(1, 28)}$=2.09, p=0.16).

These findings combine with those of *Ostlund and Balleine, (2007a)* to show that the OFC is not necessary for the flexible control of action-outcome behaviour. Furthermore, we rule out the possibility that this discrepancy between Pavlovian and instrumental devaluation effects following OFC is due to differences in the method of devaluation that is taste-aversion or specific satiety.

Using similar LO lesions we also replicate another finding reported by *Ostlund and Balleine, (2007a)*, no effect of LO lesions on sensory-specific Pavlovian-to-instrumental transfer (sPIT; *Appendix 1—figure 1*). In contrast to these relatively anterior LO lesions, more posterior OFC lesions encompassing both LO and VO have been found to disrupt the sPIT effect (*Balleine et al., 2011*; *Scarlet et al., 2012*). Chemogenetic inactivation of these posterior LO and VO aspects of the rodent OFC have also recently been found to disrupt instrumental outcome devaluation by specific satiety under some training conditions (*Parkes et al., 2018*).

## Pavlovian devaluation by taste aversion

An alternative account of the absence of OFC lesion effects on instrumental devaluation, in contrast to robust devaluation deficits by taste aversion in rodents (*Gallagher et al., 1999*; *Pickens et al., 2003*, *2005*), is the extent and specificity of OFC lesion damage. It is notable that OFC lesions in these Pavlovian devaluation studies encompass many orbital subregions (VO, LO, DLO, AI, and even MO). In contrast, the OFC lesions in the present studies, and *Ostlund and Balleine, (2007a)*, are predominantly focussed on the anterior extent of LO. In addition to testing whether these anterior LO lesions are sufficient to replicate the effect of large OFC lesions on outcome devaluation by taste aversion, a second group of lesion animals was created with posterior LO lesions. In rats, LO spans a large anterior-posterior plane (at least 3 mm), so we tested for functional heterogeneity between anterior and posterior LO subregions on Pavlovian outcome devaluation and reversal learning to identify their role in these two canonical OFC dependent tasks.

Rats underwent sham or excitotoxic lesion surgery using a range of co-ordinates, and two distinct lesion groups were established (described in Materials and methods section), anterior and posterior LO lesion groups (*Figure 2A*, *Figure 2—figure supplement 1*) were defined by damage predominantly anterior or posterior to bregma +3.70 respectively (*Figure 2B*).

First, all animals were trained on two unique Pavlovian cue-outcome relationships. Acquisition of responding to the CSs predicting the to-be devalued and non-devalued USs did not differ within groups but differed between lesion groups (*Figure 2C*) such that responding was lower in the posterior OFC lesion group. A mixed Group x CS (devalued, non-devalued) x DayBlock (4 Blocks of 3 days) ANOVA supported this observation with a significant main effect of Group ($F_{(2, 41)}$=3.67, p=0.03) and DayBlock ($F_{(3, 123)}$=102.14, p<0.001) but all other effects failed to reach significance

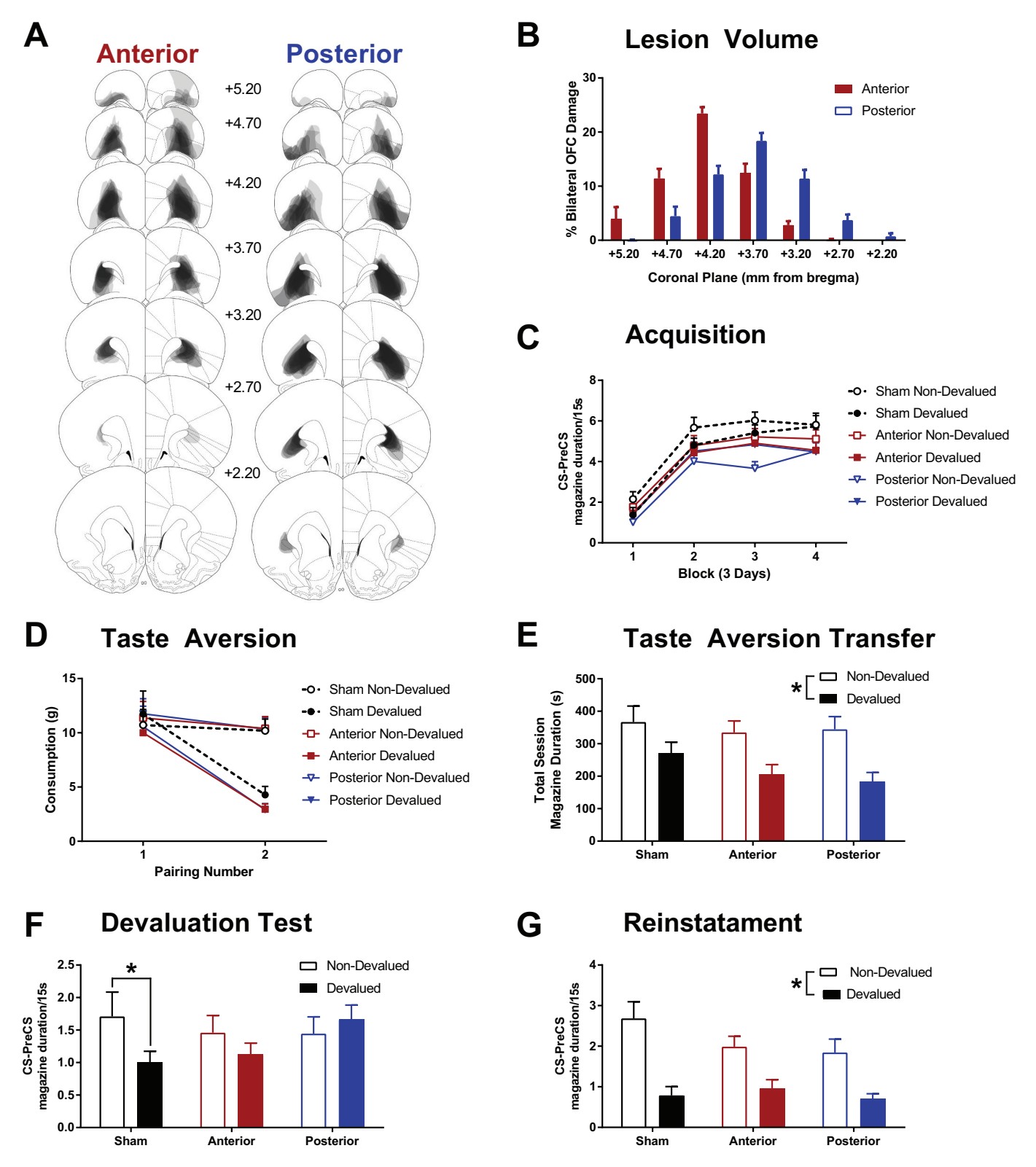

**Figure 2.** The effects of subregion specific OFC lesions on Pavlovian devaluation by taste aversion. (**A**) Representative OFC lesion damage in the anterior (left) and posterior (right) LO lesion groups; additional histology presented in *Figure 2—figure supplement 1*. Semi-transparent grey patches represent lesion damage in a single subject, and darker areas represent overlapping damage across multiple subjects. Coronal sections are identified in mm relative to bregma (Paxinos and Watson, 1997). (**B**) Quantification of percent bilateral OFC damage in anterior and posterior lesion groups at

*Figure 2 continued on next page*

*Figure 2 continued*

each coronal plane, in mm relative to bregma. (C) Rate of acquisition to the Pavlovian CSs in blocks of 3 days. Response rates presented as duration of magazine activity during the CS minus activity in the PreCS period. (D) The acquisition of a specific taste aversion following pairings of one outcome with LiCl injections (Devalued) or saline injections (Non-Devalued). The mean weight of each outcome consumed prior to each injection pairing is plotted. (E) An additional pairing of each outcome with LiCl or saline injections was conducted in the experimental chambers following non-contingent delivery of reward into magazine. Data presented as total duration of magazine activity in the session. This allowed for a measure of the transfer of the taste aversion to the testing context. (F) Magazine responding (CS – PreCS) to the CSs associated with the devalued and non-devalued outcomes, presented in extinction. (G) An outcome specific reinstatement test in which responding to the CSs was assessed after brief exposure to its associated outcome. Error bars depict + SEM. (*) Symbol denotes statistical significance of simple or main effects following a significant interaction. Effect of OFC lesions on locomotor activity presented in *Figure 2—figure supplement 2*.

DOI: https://doi.org/10.7554/eLife.37357.004

The following figure supplements are available for figure 2:

**Figure supplement 1.** Representative OFC lesion damage across the anterior-posterior plane within the same rat from the sham (left), anterior (middle), and posterior lesion groups.

DOI: https://doi.org/10.7554/eLife.37357.005

**Figure supplement 2.** OFC lesions do not affect general locomotor activity.

DOI: https://doi.org/10.7554/eLife.37357.006

(Group x US $F_{(2, 41)}$=2.55, p=0.09, Group x US x DayBlock $F_{(6, 123)}$=2.01, p=0.07, all remaining F < 1.00, p>0.44). Bonferroni corrected pairwise comparisons of overall responding revealed that the posterior group had lower performance than the sham group ($F_{(1, 41)}$=7.34, p=0.03), but no significant differences were found between anterior and sham ($F_{(1, 41)}$=2.29, p=0.42), or posterior and anterior groups ($F_{(1, 41)}$=1.65, p=0.62).

Taste aversion was successfully acquired by all groups (*Figure 2D*). Food consumption (g) was analysed using a Group x Pairing (injection 1, 2) x Devaluation (LiCl, saline) ANOVA which revealed significant effects of Devaluation ($F_{(1, 41)}$=8.23, p=0.01), Pairing ($F_{(1, 41)}$=141.39, p<0.001) and a Pairing x Devaluation interaction ($F_{(1, 41)}$=37.83, p<0.001), but no main effect or interactions with Group (all remaining F < 1.00, p>0.68). Follow up simple effects revealed that consumption of the US paired with LiCl did not differ from saline prior to the first injection (pairing 1 $F_{(1, 41)}$=0.09, p=0.77), but was significantly reduced relative to saline prior to the second injection (pairing 2 $F_{(1, 41)}$=59.199, p<0.001). The third injection pairings performed in the test chambers showed successful transfer of the taste aversion to this context in all groups (*Figure 2E*). A Group x Devaluation mixed ANOVA on magazine duration behaviour revealed a significant effect of Devaluation ($F_{(1, 41)}$=16.16, p<0.001) that did not differ with Group (all remaining F < 1.57, p>0.22). Taken together, consumption and approach towards the US paired with LiCl was successfully reduced compared to the US paired with saline injections, but the magnitude of this unique taste aversion did differ between groups.

Devaluation testing was conducted under extinction to ensure that behaviour was guided by the expected/recalled value of the outcomes (*Figure 2F*). The sham group showed a significant reduction in magazine behaviour to the CS that predicted the devalued relative to the non-devalued US, but this devaluation effect was not evident in the anterior and posterior lesion groups. This pattern of results was supported by a Group x Devaluation mixed ANOVA revealing a significant Group x Devaluation interaction ($F_{(2, 41)}$=3.46, p=0.04), the main effects of Devaluation ($F_{(1, 41)}$=3.74, p=0.06) and Group ($F_{(2, 41)}$=0.41, p=0.41) did not reach significance. Simple effects revealed that this interaction was due to a significant devaluation effect in the sham group ($F_{(1, 41)}$=7.33, p=0.01), but not the anterior ($F_{(1, 41)}$=2.06, p=0.16) or posterior groups ($F_{(1, 41)}$=0.81, p=0.37). This suggests that lesions of the anterior or the posterior LO are sufficient to disrupt Pavlovian devaluation by taste aversion, previously established with much larger OFC lesions in rodents (*Gallagher et al., 1999*; *Pickens et al., 2003*, *2005*).

Next, a US specific reinstatement test was conducted to see if the lesion groups could appropriately reduce behaviour to the devalued cue following a brief reminder of the outcome value. Rats were first exposed to one of the USs, and after a short delay they were presented with the CS that predicted that US (in extinction). All groups remained sensitive to the taste aversion when re-exposed to the USs in the test chamber (uneaten devalued USs observed by experimenter when cleaning the chamber prior to test). A mixed Group x Period (pre, post US delivery) x Devaluation ANOVA on magazine behaviour during US re-exposure (data not shown) revealed a significant effect

of Period ($F_{(1, 41)}$=71.20, p<0.001), Devaluation ($F_{(1, 41)}$=72.05, p<0.001) and Period x Devaluation interaction ($F_{(1, 41)}$=79.30, p<0.001, all remaining $F$ < 1.33, p>0.28). Simple main effects revealed that magazine behaviour did not differ before US delivery ($F_{(1, 41)}$=1.23, p=0.02), but was significantly higher after delivery of the non-devalued than the devalued US ($F_{(1, 41)}$=93.46, p<0.001).

During the reinstatement test, all groups showed significant evidence of sensitivity to US devaluation in the presence of the CSs (*Figure 2G*). A mixed Group x Devaluation ANOVA supported this pattern of results with a significant main effect of Devaluation ($F_{(1, 41)}$=50.73, p<0.001), but no significant effect of Group ($F_{(2, 41)}$=1.12, p=0.34) or Group x Devaluation interaction ($F_{(2, 41)}$=1.97, p=0.15). Therefore, re-exposure to the US prior to test elicited a robust devaluation effect in all groups. This suggests that the disruption of the Pavlovian devaluation effect following LO lesions is not caused by a failure to acquire sensory specific cue-outcome associations, not the ability to acquire a sensory specific taste-aversion, nor perseverative responding to any predictive cues. Instead, the deficit is specific to recalling the new value of the devalued outcome and/or integrating it into appropriate behavioural control.

## Sign-tracking and reversal

The finding that posterior LO lesions retarded acquisition of initial Pavlovian conditioned approach behaviour is surprising given that these animals can appropriately modulate their cue driven behaviour based on outcome value when given contact with the US in a reinstatement test. It was hypothesised that this might reflect an impairment in the attribution of value/salience to the Pavlovian cue itself. When a lever is used as a Pavlovian cue, rats will come to approach and engage with the lever cue (sign-tracking) instead of the normal conditioned approach to the magazine (goal-tracking behaviour) (*Brown and Jenkins, 1968*; *Jenkins and Moore, 1973*; *Locurto et al., 1976*). Many researchers have argued that sign-tracking behaviour reflects a process by which the lever CS acquires enhanced incentive salience so that the incentive motivational value of the outcome becomes attributed to the cue (*Berridge, 2004*). Therefore, it was predicted that the posterior LO group would not attribute incentive salience to a lever cue and show a deficit in sign-tracking. The sham, anterior, and posterior LO lesion groups were retrained on a discriminated sign-tracking procedure in using rewarded (CS+) and non-rewarded (CS-) lever cues (left and right lever, counterbalanced).

To ensure that any differences in lever pressing are not confounded by differential levels of competing responses, it is important to establish that there are no group differences in baseline magazine behaviour. Mixed Group x DayBlock (4 blocks of 3 days) ANOVAs for the PreCS magazine duration did not differ between groups during acquisition (Group or Group x DayBlock interactions, all $F$ < 1.75, p>0.12) or subsequent reversal (all $F$ < 2.01, p>0.14, data not shown).

During acquisition, lever pressing during the CS+ was greater than CS-, but the lesion groups made fewer responses than the sham group (*Figure 3A*, left panel). A mixed Group x CS (CS+, CS-) x DayBlock (4 blocks of 3 days) ANOVA partially supported the observed differences with a significant main effect of Group ($F_{(2, 41)}$=3.75, p=0.03) and a 3-way Group x CS x DayBlock interaction ($F_{(6, 123)}$=3.42, p<0.01, all remaining effects also reached significance $F$ > 2.20, p<0.05). While there were no group differences on DayBlocks 1 and 2 (non-significant main effects of Group and Group x CS interactions for DayBlock 1 and 2, all $F$ < 2.27, p>0.12), on DayBlocks 3 and 4 there were significant main effects of Group (DayBlock 3 $F_{(2, 41)}$=4.97, p=0.01, DayBlock 4 $F_{(2, 41)}$=5.01, p=0.01) and Group x Cue interactions (DayBlock 3 $F_{(2, 41)}$=3.99, p=0.03, DayBlock 4 $F_{(2, 41)}$=4.70, p=0.01). Bonferroni corrected simple effects revealed that there were no group differences in responding to the CS- (DayBlock 3 and 4, all $F$ < 4.21, p>0.14), whereas CS +lever pressing was greater in the sham than the posterior group (DayBlock 3 $F_{(1, 41)}$=9.51, p=0.01, DayBlock 4 $F_{(1, 41)}$=8.77, p=0.02) but not different between sham and anterior or anterior and posterior lesions (DayBlock 3 and 4, all $F$ < 3.87, p>0.17). Therefore, lever responding to the CS+ was lower for the posterior lesion than the sham group in the second half of acquisition but no differences between anterior lesions and the sham or posterior lesion groups were revealed.

Magazine duration responding in the CS- decreased across acquisition in all groups whereas responding to the CS+ only decreased in the sham and anterior groups but not in the posterior lesion group (*Figure 3A*, left panel). A mixed Group x CS x DayBlock ANOVA supported these observations. CS+ responding was greater than CS- responding, and while responding decreased across days this decline was more rapid to the CS- than the CS+ (main effect of DayBlock $F_{(3,}$

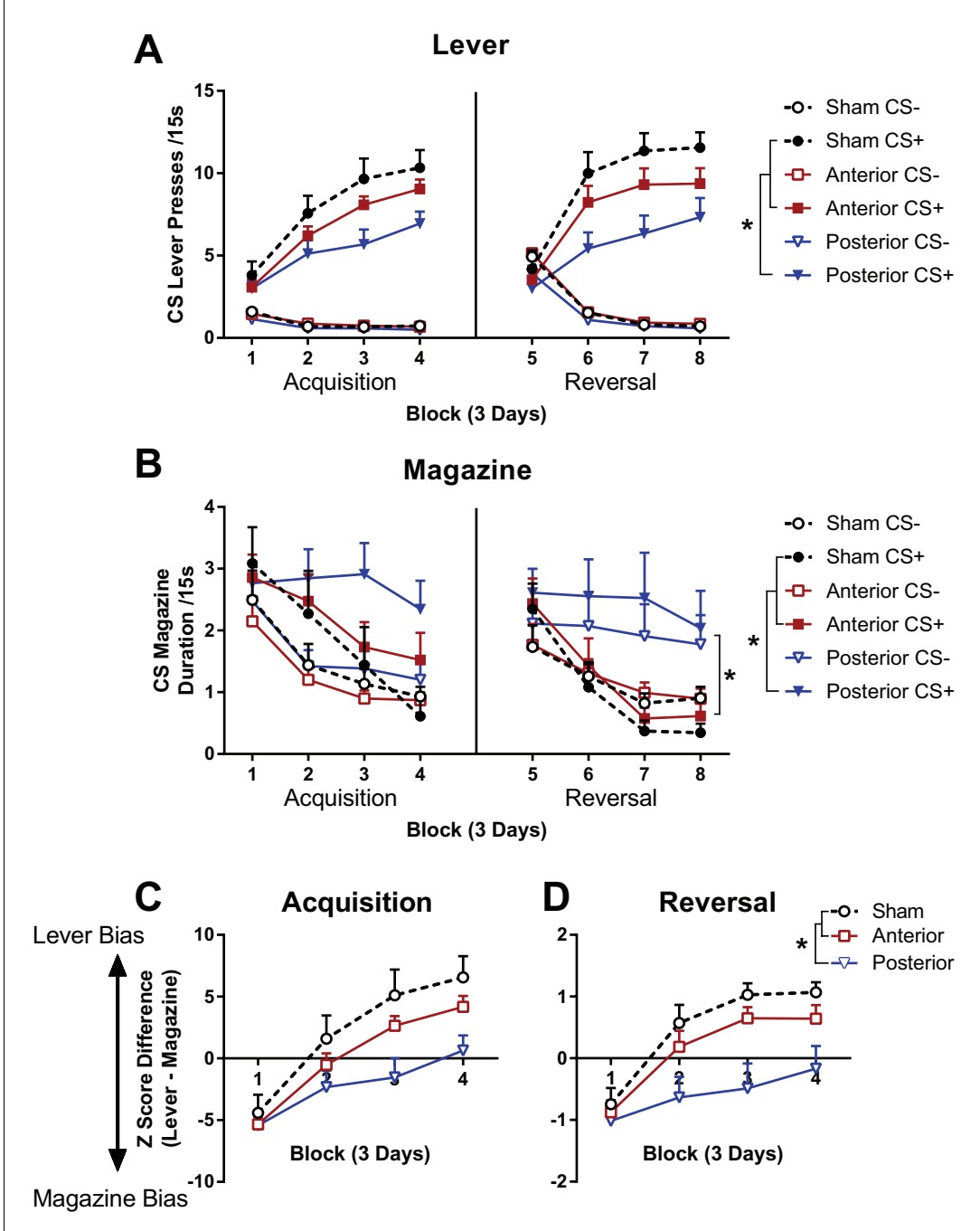

**Figure 3.** The effects of subregion specific OFC lesions sign-tracking behaviour and reversal learning. Lever (**A**) and magazine (**B**) CS responding during 12 days of acquisition (left), and reversal (right) of a rewarded (CS+) and non-rewarded (CS-) lever cue. Response competition during acquisition (**C**) and reversal (**D**) of the CS+. Lever response bias calculated as the difference between standardised lever and magazine responding so that positive scores represent greater lever bias and negative scores represent greater magazine response bias. Error bars depict + SEM. (*) Symbol denotes statistical significance of simple or main effects following a significant interaction.

DOI: https://doi.org/10.7554/eLife.37357.007

$_{123)}$=30.06, p<0.001, and a CS x DayBlock interaction $F_{(3, 123)}$=5.82, p=0.001). A 3-way Group x CS x DayBlock interaction ($F_{(6, 123)}$=2.94, p=0.01, and a significant Group x DayBlock interaction $F_{(3, 123)}$=2.19, p<0.05) suggested that the differential decline in responding to each CS was not the same in each group (all remaining $F < 1.00$, p>0.39). Separate follow-up Group x DayBlock ANOVAs

were conducted on each CS to explore the 3-way interaction. Responding during the CS- decreased to the same extent in all groups (significant main effect of DayBlock $F_{(3, 123)}$=32.25, p<0.001, but no main effect of Group $F_{(2, 41)}$=0.55, p=0.58, or Group x DayBlock interaction $F_{(3, 123)}$=0.24, p=0.96). In contrast, during the CS+ there were significant group differences in responding (significant main effect of DayBlock $F_{(3, 123)}$=18.23, p<0.001, no main effect of Group $F_{(2, 41)}$=1.06, p=0.26, significant Group x DayBlock interaction $F_{(3, 123)}$=3.65, p<0.01). Simple main effects analysing group differences in CS+ magazine duration found that that there were no group differences on DayBlocks 1 and 2 and 3 (effect of Group on DayBlock 1 $F_{(2, 41)}$=0.14, p=0.87, DayBlock 2 $F_{(2, 41)}$=0.30, p=0.75, Day-Block 3 $F_{(2, 41)}$=2.38, p=0.11) however there were significant group differences on DayBlock 4 (effect of Group on DayBlock 4 $F_{(2, 41)}$=4.81, p=0.01). Follow up Bonferroni corrected comparisons revealed that on DayBlock 4, posterior group magazine responding was greater than the sham group ($F_{(1, 41)}$=8.12, p=0.02).

A popular measure of sign-tracking behaviour is the Pavlovian conditioned approach (PCA) index (*Flagel et al., 2011*) which combines a number measures relating to the probability, latency, and relative bias in lever pressing over magazine approach. However, there is no principled justification for the specific choice or relative weighting of these measures, so a data driven alternative was used to quantify sign-tracking behaviour. A Doubly Multivariate ANOVA (*Tabachnick and Fidell, 2013*) was employed to directly assess response competition between the lever pressing and magazine duration measures in the sign-tracking procedure. This allowed for the comparison of two fundamentally different measures, lever pressing and magazine duration, which are likely to be correlated due to response competition that is high scores on one measure preclude high scores on the other measure.

A Group x DayBlock x CS (rewarded, non-rewarded) MANOVA with lever pressing and magazine duration response measures revealed a significant Group x DayBlock x CS interaction ($F_{(6, 37)}$=2.95, p=0.02). Follow up Group x DayBlock MANOVAs revealed a significant Group x DayBlock interaction for the rewarded CS ($F_{(6, 37)}$=2.59, p=0.03) but not the non-rewarded CS ($F_{(6, 37)}$=0.86, p=0.53). This significant multivariate interaction was investigated using a planned composite of the differences between measures (standardised with respect to within group variances and the grand mean) that is lever pressing - magazine duration. This composite reflects the expected competition between responses and was verified (post-hoc) by a discriminant analysis on the final day block of acquisition which revealed standardised coefficients of 0.64 (lever pressing) and −0.56 (magazine duration) associated with the first eigenvalue.

The difference scores on the standardised variate revealed that during acquisition of responding directed at the rewarded lever, all groups expressed a bias towards magazine responding at the start of training (*Figure 3C*). However, by the end of training the sham and anterior groups were responding more to the lever than the magazine whereas the posterior group was responding similarly to both the magazine and the lever. This pattern of observed results was supported statistically. A Group x DayBlock ANOVA on the acquisition of the rewarded lever revealed a significant main effect of Group ($F_{(2, 41)}$=3.45, p=0.04), DayBlock ($F_{(3, 123)}$=95.89, p<0.001) and Group x DayBlock interaction ($F_{(6, 123)}$=3.67, p<0.01). Bonferroni adjusted simple effects revealed that there were no group differences in on DayBlock 1 and 2 (all p>0.17) whereas on DayBlock 3 and 4 the posterior group had significantly lower scores than the sham group (p<0.01, all remaining p>0.15). These findings suggest that there is a difference in OFC function within LO along the anterior-posterior gradient.

## Reversal learning

A commonly reported deficit following OFC damage is in reversal learning (*Boulougouris et al., 2007*; *Rudebeck and Murray, 2011a*; *Schoenbaum et al., 2002*), so a reversal manipulation was employed to test whether anterior and posterior LO damage result in a reversal deficit. The identity of the CS+ and CS- levers was reversed and acquisition continued for another 12 days.

Reversal learning resulted in more lever presses being directed towards the new CS+ than the CS-, but the lesion groups made fewer responses than the sham group (*Figure 3A*, right panel). A mixed Group x CS (CS+, CS-) x DayBlock (4 blocks of 3 days) ANOVA partially supported the observed differences with a significant main effect of Group ($F_{(2, 41)}$=5.09, p=0.01) and a 3-way Group x CS x DayBlock interaction ($F_{(6, 123)}$=3.94, p=0.001, all remaining effects also reached significance $F > 3.56$, p<0.04, except the Group x DayBlock interaction $F_{(6, 123)}$=1.52, p=0.181). The 3-way

interaction was decomposed into separate Group x CS ANOVAs conducted for each DayBlock. On DayBlock 1 responding was greater to the CS- than the CS+ (main effect of CS on DayBlock 1 $F_{(1, 41)}$=5.27, p=0.03) but this did not differ between groups (non-significant main effect of Group and Group x CS interaction, all $F < 1.82$, p>0.18). On DayBlocks 2, 3 and 4 the main effect of CS remained significant such that CS+ responding was now higher than CS- responding (DayBlock 2 $F_{(1, 41)}$=109.59, p<0.001, DayBlock 3 $F_{(1, 41)}$=185.42, p<0.001, DayBlock 4 $F_{(1, 41)}$=222.47, p<0.001), however there were also significant main effects of Group (DayBlock 2 $F_{(2, 41)}$=5.05, p=0.01, DayBlock 3 $F_{(2, 41)}$=5.42, p=0.01, DayBlock 4 $F_{(2, 41)}$=4.09, p=0.02) and Group x Cue interactions (DayBlock 2 $F_{(2, 41)}$=3.60, p=0.04, DayBlock 3 $F_{(2, 41)}$=5.45, p=0.01, DayBlock 4 $F_{(2, 41)}$=3.97, p=0.03). Bonferroni corrected simple effects revealed that there were no group differences in responding to the CS- (DayBlock 2, 3 and 4, all $F < 5.01$, p>0.09), whereas CS+ lever pressing was greater in the sham than the posterior group (DayBlock 2 $F_{(1, 41)}$=8.47, p=0.02, DayBlock 3 $F_{(1, 41)}$=10.72, p=0.01, DayBlock 4 $F_{(1, 41)}$=8.10, p=0.02) but not different between sham and anterior, or anterior and posterior lesions (DayBlock 2, 3 and 4, all $F < 4.16$, p>0.14). Therefore, similar to initial acquisition, lever responding to the CS+ was lower for the posterior lesion than the sham group later in acquisition but no differences between anterior lesions and the sham or posterior lesion groups could be concluded.

A reversal deficit was also found in the posterior LO lesion group using the measure of magazine duration. Magazine duration decreased more rapidly to the CS+ than the CS- during reversal for the sham and anterior lesion group but there was no apparent reduction in responding to either CS in the posterior lesion group (*Figure 3B*, right panel). These observations were supported by a Group x CS x DayBlock ANOVA on magazine duration data which revealed significant differences in responding to each CS (CS x DayBlock interaction $F_{(3, 123)}$=5.24, p<0.01) and group differences in the rate of response reduction across the session (main effect of Group $F_{(2, 41)}$=3.44, p=0.04, main effect of DayBlock $F_{(3, 123)}$=18.48, p<0.001, and Group x DayBlock interaction $F_{(6, 123)}$=2.43, p=0.03). Overall, simple main effects revealed that responding was higher for CS+ than CS-on DayBlock 5 ($F_{(1, 41)}$=17.38, p<0.001), but at similar levels on DayBlock 6, 7 and 8 (all $F_{(1, 41)}<1.00$, p>0.28). Simple main effects examining group differences revealed that responding reduced across reversal in the sham and anterior lesion groups (effect of DayBlock for sham group $F_{(3, 39)}$=5.30, p<0.01, anterior group $F_{(3, 39)}$=4.69, p=0.01) but not in the posterior lesion group ($F_{(3, 39)}$=2.33, p=0.09).

Similar to the analysis of acquisition, a multivariate approach was used to assess competition between lever and magazine behaviour in reversal. A Group x DayBlock x CS (rewarded, non-rewarded) MANOVA with lever pressing and magazine duration response measures revealed a significant Group x DayBlock x CS interaction ($F_{(6, 37)}$=3.11, p=0.01). Follow up Group x DayBlock MANOVAs revealed a significant Group x DayBlock interaction for the rewarded CS ($F_{(6, 37)}$=3.34, p=0.01) but not the non-rewarded CS ($F_{(6, 37)}$=2.11, p=0.08). A discriminant analysis was performed on the final day block of acquisition which revealed standardised coefficients of 0.41 (lever pressing) and −0.69 (magazine duration) associated with the first eigenvalue, which supported the choice of a difference score again.

The difference scores on the standardised variate revealed that during reversal of the rewarded lever, all groups responded more towards the magazine than the lever at the start of training (*Figure 3D*). However, by the end of training the sham and anterior groups were performing more to the lever than the magazine whereas the posterior group was performing equally to both the magazine and the lever. A Group x DayBlock ANOVA on the acquisition of the rewarded lever revealed a significant main effect of Group ($F_{(2, 41)}$=5.30, p=0.01), DayBlock ($F_{(3, 123)}$=47.55, p<0.001) and Group x DayBlock interaction ($F_{(6, 123)}$=3.20, p=0.01). Bonferroni adjusted simple effects revealed that there were no group differences in on DayBlock 1 (all p>0.99) whereas on Day-Blocks 3 and 4 the posterior group had significantly lower scores than the sham group (all p<0.01, all remaining p>0.05). Similar to acquisition, the posterior lesion group were significantly impaired in sign-tracking to the CS+ during reversal.

## Discussion

Our results demonstrated two important neural and behavioural dissociations within the rodent OFC. First, we directly confirmed the dissociable role of the rodent OFC in Pavlovian but not instrumental behavioural flexibility following outcome devaluation (*Gallagher et al., 1999*; *Ostlund and*

*Balleine, 2007a*). Second, we showed a novel dissociation within anterior and posterior subregions of rodent LO in outcome devaluation, sign-tracking, and reversal learning procedures. Together, these findings indicate that many contradictory findings in OFC research may be reconciled as functional heterogeneity within the putative orbital subregions.

## Outcome devaluation

Successfully updating behaviour in an outcome devaluation procedure provides strong evidence that the organism has (i) a representation of the specific identity of the predicted outcome, (ii) access to its current motivational value, and (iii) can flexibly update behaviour based on this information. Prominent model-based and sensory-specific outcome-expectancy coding accounts of the OFC argue that deficits in outcome devaluation following OFC lesions are due to an inability to access the representation of the specific identity of expected outcomes (*Delamater, 2007*; *Wikenheiser et al., 2017*; *Wilson et al., 2014*). Alternatively, these deficits can be modelled as an inability to use a cognitive model of the task structure to mentally 'simulate' the consequences of their actions on future states (*Wilson et al., 2014*).

Model-based and sensory-specific outcome-expectancy coding accounts of the OFC (*Delamater, 2007*; *Rudebeck and Murray, 2014*; *Wilson et al., 2014*) predict that OFC lesions should disrupt the devaluation effect in both instrumental and Pavlovian outcome devaluation. However, the absence of an effect of OFC lesions on instrumental devaluation suggests that the representation of the specific properties of instrumental and Pavlovian outcomes are dissociable (*Ostlund and Balleine, 2007b*). This absence also suggests that organisms represent task states and/or state transitions differently if they are caused by an instrumental action or an external Pavlovian stimulus, a distinction that has recently been incorporated into some model-based reinforcement learning theories (*Dayan and Berridge, 2014*; *Lesaint et al., 2014*; *Zhang et al., 2009*). Consistent with a selective role for the OFC in Pavlovian model-based inferences, there is mounting evidence that the OFC is necessary for making inferences in procedures that require a model-based representation the relationship between external cues (*Jones et al., 2012*; *Sadacca et al., 2018*). Given that outcome devaluation is considered a canonical deficit following OFC lesions, the absence of a lesion deficit in instrumental outcome devaluation must be incorporated into theories of OFC function.

In this paper we explored a number of possible reasons for the reported absence of an OFC lesion deficit in instrumental outcome devaluation (*Ostlund and Balleine, 2007a*). One possibility is that the authors used task parameters that were not appropriate to detect a subtle OFC lesion deficit. For example, the number of distinct responses/outcomes that are trained concurrently affects the sensitivity of instrumental conditioning to outcome devaluation. Behaviour can become insensitive to devaluation (habitual) with overtraining (*Adams, 1982*; *Dickinson, 1985*) if only a single lever-outcome procedure is used. This overtraining effect is abolished if training procedures allow the animal to experience multiple distinct action-outcome contingencies (*Colwill and Rescorla, 1985*; *Kosaki and Dickinson, 2010*). *Ostlund and Balleine, (2007a)* employed two unique levers and outcomes during acquisition, and a simultaneous two-lever choice test following devaluation. We tested whether these parameters, which have been shown to promote devaluation sensitive behaviour, masked a subtle OFC lesion deficit in instrumental devaluation. Our task employed a single lever-outcome design, and a random interval schedule of training, both of which have been shown to encourage the formation of habitual/devaluation insensitive behaviour (*Adams, 1982*; *Dickinson, 1985*). Despite these parameters, both sham and lesion groups exhibited a weak but robust devaluation effect (*Figure 1D*).

Another possibility is that the method of devaluation can affect whether OFC lesion deficits are observed in instrumental outcome devaluation. *Ostlund and Balleine, (2007a)* used sensory specific satiety rather than lithium-chloride induced taste aversion as the method of devaluation. These two devaluation methods are often used interchangeably in computational models of learning (e.g. *Dranias et al., 2008*; *Grossberg et al., 2008*) and often yield similar results following lesions of neural regions involved in goal-directed and habitual behavioural control (*Coutureau and Killcross, 2003*; *Killcross and Blundell, 2002*; *Killcross and Coutureau, 2003*). Our results confirm that the absence of an OFC lesion deficit on instrumental devaluation is not simply due to a difference between lithium-chloride taste aversion and sensory specific satiety devaluation methods.

In contrast to instrumental devaluation, we found that OFC lesions abolished the appropriate reduction in Pavlovian approach responding when the outcome was devalued by taste aversion

(*Gallagher et al., 1999*; *Pickens et al., 2003*, *2005*). These findings confirm that focal lesions of LO are sufficient to disrupt Pavlovian devaluation in rodents, previously reported with substantially larger OFC lesions targeting VO, LO, DLO, AI, and MO subregions (*Gallagher et al., 1999*; *Pickens et al., 2003*, *2005*).

Consistent with model-based theories of OFC function, at test LO lesions disrupted the ability to infer the new expected value of the CS that predicted the devalued outcome based. Specifically, OFC is argued to be necessary for inferring 'hidden' task states, such as the new value of the expected outcome, when this information is not externally available/signalled (*Wilson et al., 2014*). Consistent with this prediction, providing a brief re-exposure to the devalued reward immediately prior to test (reinstatement test) allowed LO lesioned animals to successfully integrate the devalued outcome into their anticipatory responding. However, this result must be interpreted with caution as it is possible that re-exposure to the devalued US in the magazine resulted in some form of short-term avoidance to the magazine that persisted throughout the subsequent test session when the devalued CS was presented.

## Sign-tracking and reversal learning

A consistently reported finding is that OFC lesions leave intact the initial acquisition and behavioural expression of either cue-outcome or action-outcome contingencies (*Boulougouris et al., 2007*; *Chudasama et al., 2007*; *Chudasama and Robbins, 2003*; *Gallagher et al., 1999*; *Rudebeck and Murray, 2011a*; *Schoenbaum et al., 2002*; but see *Walton et al., 2011*). This finding is critical to ruling out many alternative explanations of the effect of OFC lesions on outcome devaluation such as general learning deficits. In the present experiments, anterior LO lesions did not affect instrumental conditioning, Pavlovian conditioning, or taste aversion learning. Unexpectedly, lesions that were focused on the posterior LO region did suppress behavioural responding during Pavlovian acquisition. This effect is not simply a general suppression of activity, as there was no difference in locomotor activity (*Figure 2—figure supplement 2*), nor a change in appetite, as there was no difference in consumption levels at the start of taste aversion learning.

One possible account of the reduced Pavlovian conditioned approach behaviour in the posterior LO group is that the CS did not acquire incentive salience. Incentive salience refers to the process by which the incentive-motivational properties of the outcome are transferred to the CS (*Berridge, 2004*), such that if a lever CS is presented a rat will attempt to 'consume' the lever as if it were the pellet that it predicts. This behaviour directed at the lever CS (sign-tracking) comes at the expense of the traditional Pavlovian approach response to the site of reward delivery, the magazine (goal-tracking). Sham control and anterior LO lesions did not affect the propensity to acquire sign-tracking behaviour, whereas sign-tracking was significantly reduced following posterior LO lesions. This finding is consistent with evidence that rats showing stronger sign-tracking tendencies have increased c-fos activity in posterior OFC regions following lever cue presentation (*Flagel et al., 2011*). This suggests that the posterior but not the anterior LO mediates the attribution of incentive-salience to cues. Alternatively, posterior LO may be involved in resolving response competition when multiple responses are supported by a predictive cue. In the present experiment, extensive Pavlovian training during the outcome devaluation procedure preceded the sign-tracking procedure, which may have resulted in a pre-existing dominant magazine approach response that could not be overcome following posterior LO lesions.

Surprisingly, extensive LO lesions have also been shown to have no effect on sign-tracking behaviour (*Chang, 2014*), but did retard subsequent reversal learning when rewarded (CS+) and non-rewarded (CS-) lever cues reversed reward contingencies. The present study found a similar impairment in reversal learning following posterior but not anterior LO lesions. This reversal deficit was not simply due to differences in the acquisition of the sign-tracking response as the posterior LO lesioned animals could reverse their lever approach behaviour. Instead, the deficit was specific to the magazine approach response which failed to extinguish in the posterior LO group when the previously rewarded CS+ was reversed to a non-rewarded CS-.

It is important to consider the limitations of a pre-training lesion approach to manipulating OFC function. While pre-training lesions guarantee loss of OFC function throughout training, it is possible that other neural regions might compensate for the loss of this function. Pre-training lesions are an important approach for probing deficits in acquisition, but limit inferences about whether these deficits reflect impaired encoding or retrieval. For example, ehile it has been shown that OFC lesions

disrupt outcome devaluation when performed before and after training (*Gallagher et al., 1999*; *Pickens et al., 2003*, *2005*), the effect of OFC lesions on reversal learning depends on whether they occur before or after initial training (*Boulougouris et al., 2007*; *Boulougouris and Robbins, 2009*). Further research is required to clarify the nature and extent of the anterior and posterior LO lesion deficits and how they relate to the function of the orbitofrontal region as a whole.

## Rodent and primate homology

Human and non-human primate OFC can be defined cytoarchitectonically using clear granular, agranular, and dysgranular areas (*Price, 2006*). In contrast, the rodent OFC only consists of agranular cortical regions, which led *Brodmann, 1909* to conclude that rodents do not have a comparable orbital or frontal cortex. However, *Rose and Woolsey, 1948* proposed a different approach to identifying rodent homologs of the orbital and frontal cortex based on similar connectivity between the putative OFC of rabbits and cats and the mediodorsal nucleus of the thalamus (MD). This approach, based on MD connectivity, has been repeatedly extended to other regions of the frontal cortex in rodents (*Groenewegen, 1988*; *Uylings et al., 2003*). However, *Price, 2006* noted that Brodmann's original problem of defining precise homologs between rodent and primate OFC with comparable cytoarchitecture still remains (an argument that some researchers have maintained, for example *Preuss, 1995*; *Rolls, 2014*; *Rudebeck and Murray, 2011a*; *Wise, 2008*). The solution to this problem has been to base rodent and primate OFC homology on a combination of similar connectivity and functional evidence (e.g. *Roesch and Schoenbaum, 2006*; *Rudebeck and Murray, 2014*).

The hallmark behavioural consequences of OFC lesions in rodents and primates, critical to establishing cross-species homology, have been questioned. For example, deficits in extinction learning have been cited and form the basis of models of OFC function (*Butter, 1969*; *Kolb et al., 1974*; *Wilson et al., 2014*), but have been poorly replicated (*Burke et al., 2009*; *Panayi and Killcross, 2014*). The two behavioural disturbances following OFC damage that have dominated the literature (*Murray et al., 2007*) are impaired reversal learning and outcome devaluation deficits. Recently, the robustness of reversal learning deficits following OFC lesions in primates has been challenged as an artefact of aspiration lesions which can cause unintended damage to neighbouring white matter tracts (*Rudebeck et al., 2013*). Fibre sparing excitotoxic lesions fail to replicate reversal learning deficits but do significantly disrupt outcome devaluation in primates. This finding has important implications for questions of homology between primate and rodent OFC as it suggests very few functional similarities exist. However, the apparent lack of functional similarities may be a consequence of poor OFC subregion specificity within the rodent and primate literature.

Our findings provide the first evidence of a dissociation of devaluation and reversal learning deficits within anterior and posterior regions of the lateral OFC subregion. Specifically, both anterior and posterior LO are necessary for updating behaviour based on the current value of expected outcomes (i.e. disrupt devaluation performance), but only posterior LO appears to be necessary for rapidly updating behaviour when predictive cue-outcome contingencies change (i.e. reversal learning deficits). Recently *Murray et al. (2015)* provided similar demonstrations of functional dissociations between anterior (area 11) and posterior (area 13) macaque OFC in Pavlovian outcome devaluation. Together these data suggest the importance of anterior-posterior differences in OFC subregions that complement the growing literature on functional differences between medial and lateral OFC subregions in both rodents (*Balleine et al., 2011*; *Bradfield et al., 2015*; *Corwin et al., 1994*; *Izquierdo, 2017*; *Mar et al., 2011*) and primates (*Bouret and Richmond, 2010*; *Noonan et al., 2010*; *Rudebeck and Murray, 2011a*; *Walton et al., 2015*).

## Theoretical accounts of OFC function

The importance of differentiating OFC subregions has implications for theories of OFC function. One class of theories of OFC function proposes that the OFC represents information about the sensory specific properties of expected outcomes (*Burke et al., 2008*; *Delamater, 2007*; *Schoenbaum and Esber, 2010*; *Schoenbaum et al., 2009*). During Pavlovian conditioning in normal animals, a stimulus may form associations with multiple features of a reward such as its general motivational properties and sensory specific properties; Associative activation of these different properties can lead to different classes of responses such as general preparatory or specific consummatory responses (*Delamater, 2007*, *2012*; *Dickinson and Dearing, 1979*; *Hall, 2002*; *Konorski, 1967*;

*Wagner and Brandon, 1989*). Here, the OFC is proposed as the neural substrate of the associatively activated representation of an expected reward's sensory specific properties. For example, if an animal learns that a tone stimulus predicts lemon flavoured sucrose reward, then in the presence of the tone and in anticipation of reward delivery the OFC might represent information about the lemon flavour, viscous fluid properties, and sweet taste of the upcoming reward. This theory accounts for the effect of OFC lesions in outcome devaluation since an animal needs to know which outcome is predicted (outcome identity) to selectively reduce anticipatory responding for a no-longer valuable outcome.

Model-based theories of OFC function can be considered modern extensions of these sensory-specific encoding accounts. When the sensory-sensory associations formed between a CS and the sensory properties of a reward are relevant to solving a task (as in outcome devaluation), they can be interpreted more generally as forming part of the task structure. We propose that our lesioned animals could represent the specific properties of the expected outcome but could not use this representation/underlying task-structure to access the current motivational value of that outcome. This proposal is in line with sensory specific outcome expectancy theories of OFC function but suggests a limited role for the anterior LO in accessing the current/updated expected value of an outcome based on its sensory properties and using this to modulate behaviour accordingly. It may be that a unified representation of an expected outcome, such as predicted likelihood, taste, location, hedonic value, and motivational value is represented across multiple OFC subregions.

Further evidence for the distribution of these representations across multiple OFC subregions in rodents comes from recent studies showing similar dissociations. For example, we replicated the absence of an effect of anterior LO lesions on sensory specific Pavlovian-to-instrumental transfer (sPIT, *Appendix 1—figure 1A*; *Ostlund and Balleine, 2007a*), a procedure that requires inferences between a number of hidden task states. In contrast, larger pre-training lesions encompassing both LO and VO do disrupt the sPIT effect (*Balleine et al., 2011*; *Scarlet et al., 2012*), but not instrumental outcome devaluation. Recently, *Parkes et al. (2018)* showed that chemogenetic disruption of posterior LO and DLO can disrupt instrumental devaluation under certain training conditions. Together, these findings suggest that the role of the OFC as a whole may be to generate a cognitive map of underlying task structure. However, the encoding and subsequent use of these underlying task structures to guide behaviour appears to be distributed amongst the many orbital subregions.

A similar conclusion is reached by *Murray et al. (2015)* in macaques, who found that temporary inactivation of anterior OFC (area 11) disrupted satiety devaluation when inactivation occurred at test but not when inactivation occurred during the satiety devaluation procedure prior to test. In contrast, posterior OFC (area 13) inactivation only disrupted performance when inactivated during the satiety procedure but not at test. This suggests that posterior OFC in macaques is necessary for updating the value of expected rewards, whereas anterior OFC is critical for translating this knowledge into behaviour. This parallels our suggested role for the anterior LO in rodents in accessing the current value of an expected outcome to guide behaviour. Furthermore, this potential homology predicts that posterior LO in rodents might be important for value updating. Our findings provide prima facie evidence for this prediction, showing that posterior LO lesions suppress overall levels of Pavlovian learning, and extinction of learnt value during reversal learning, consistent with impoverished value updating. However, direct tests of this dissociation are still needed to confirm this homology between rodent and non-human primate OFC.

## Materials and methods

### Animals

Rats were housed four per cage in ventilated Plexiglass cages in a temperature regulated (22 ± 1°C) and light regulated (12 hr light/dark cycle, lights on at 7:00 AM) colony room. At least one week prior to behavioural testing, feeding was restricted to ensure that weight was approximately 95% of ad libitum feeding weight, and never dropped below 85%. All animal research was carried out in accordance with the National Institute of Health Guide for the Care and Use of Laboratories Animals (NIH publications No. 80–23, revised 1996) and approved by the University of New South Wales Animal Care and Ethics Committee. Subjects were forty-eight male Long Evans rats (Monash Animal Services, Gippsland, Victoria, Australia) approximately 4 months old (Experiment 1, N = 32, weighing

between 301–359 g, M = 326.6 g; Appendix 1 Experiment, N = 16, weighing between 321–399 g, M = 342.1 g), and one hundred and twelve male Wistar rats (BRC Laboratory Animal Service, University of Adelaide, South Australia, Australia) approximately 4 months old (Experiment 2, N = 64, weighing between 343–452 g, M = 403.6 g).

## Apparatus

Behavioural testing was conducted in eight identical operant chambers (30.5 × 32.5 × 29.5 cm; Med Associates) individually housed within ventilated sound attenuating cabinets. Each chamber was fitted with a 3 W house light that was centrally located at the top of the left-hand wall. Food pellets could be delivered into a recessed magazine, centrally located at the bottom of the right-hand wall. Delivery of up to two separate liquid rewards via rubber tubing into the magazine was achieved using peristaltic pumps located above the testing chamber. The top of the magazine contained a white LED light that could serve as a visual stimulus. Access to the magazine was measured by infrared detectors at the mouth of the recess. Two retractable levers were located on either side of the magazine on the right-hand wall. A speaker located to the right of the house light could provide auditory stimuli to the chamber. In addition, a 5 Hz train of clicks produced by a heavy-duty relay placed outside the chamber at the back right corner of the cabinet was used as an auditory stimulus. The chambers were wiped down with ethanol (80% v/v) between each session. A computer equipped with Med-PC software (Med Associates Inc., St. Albans, VT, USA) was used to control the experimental procedures and record data.

### Devaluation chambers.

To provide individual access to reinforcers during the devaluation procedure, rats were individually placed into a mouse cage (33 × 18 × 14 cm clear Perspex cage with a wireframe top). Pellet reinforcers were presented in small glass ramekins inside the box and liquid reinforcers were presented in water bottles with a sipper tube. 1 day prior to the start of the devaluation period, all rats were exposed to the mouse cages and given 30 mins of free access to home cage food and water to reduce novelty to the context and consuming from the ramekin and water bottles.

Locomotor activity was assessed in a set of 4 rat open field arenas (Med Associates Inc., St. Albans, VT, USA) individually housed in light and sound attenuating cabinets. A 3 W light attached on the top left corner of the sound attenuating cabinet provided general illumination in the chamber and was always on. A 28 V DC fan on the right hand wall of the sound attenuating cabinet was also left on throughout testing to mask outside noise. The floor of the open field arena was smooth plastic and the four walls were clear Perspex with a clear Perspex roof containing ventilation holes. The internal dimensions of the chamber were 43.2 × 43.2 × 30.5 cm (length x width x height). Two opposing walls contained an array of 16 evenly spaced infrared detectors set 3 cm above the floor to detect animal locomotor activity. A second pair of infrared beam arrays was set 14 cm above the floor on the remaining walls to detect rearing behaviours. Infrared beam breaks were recorded using a computer equipped with Activity Monitor software (Med Associates Inc., St. Albans, VT, USA) which provided a measure of average distance travelled based on beam break information.

## Surgery

Excitotoxic lesions targeting the lateral OFC were performed prior to any training. Rats were anesthetized with isoflurane, their heads shaved, and placed in a stereotaxic frame (World Precision Instruments, Inc., Sarasota, FL, USA). The scalp was incised, and the skull exposed and adjusted to flat skull position. Two small holes were drilled into the skull and the dura mater was severed to reveal the underlying cortical parenchyma. A 1 μL Hamilton needle (Hamilton Company, Reno, NV, USA) was lowered through the two holes targeting the lateral OFC (co-ordinates specified below). At each site the needle was first left to rest for 1 min. Then an infusion of N-methyl-D-aspartic acid (NMDA; Sigma-Aldrich, Switzerland), dissolved in phosphate buffered saline (pH 7.4) to achieve a concentration of 10 μg/μL, was infused for 3 mins at a rate of 0.1 μ/min. Finally, the needle was left in situ for a further 4 mins to allow the solution to diffuse into the tissue. Following the diffusion period the syringe was extracted and the scalp cleaned and sutured. Sham lesions proceeded identically to excitotoxic lesions except that no drugs were infused during the infusion period. After a

minimum of 1 week of postoperative recovery, rats were returned to food restriction for 2 days prior to further training.

Animals were randomly assigned to one of two lesion conditions in Experiments 1and Supplementary Experiment (*Appendix 1—figure 1A*), with the following stereotaxic co-ordinates AP: +3.5 mm, ML: ±2.2 mm, D-V: −5.0 mm from bregma (Experiment 1, sham, n = 16; lesion, n = 16; Supplementary Experiment, sham, n = 8; lesion, n = 8). In Experiment 2, three sets of lesion co-ordinates were used to encourage distinct lesion subgroups. The co-ordinates used were AP: +4.2 mm, ML: ±2.6 mm, D-V: 4.8 mm (n = 16 lesion, n = 6 sham), AP: +3.7 mm, ML: ±3.2 mm, D-V: −5.0 mm (n = 16 lesion, n = 5 sham) and AP: +3.7 mm, ML: ±2.6 mm, D-V: −5.0 mm (n = 16 lesion, n = 5 sham). Final group designation was based on post-experimental lesion characterisation.

## Reinforcers

The reinforcers used were a single grain pellet (45 mg dustless precision grain-based pellets; Bioserv, Frenchtown, NJ, USA), 20% w/v sucrose solution and 20% w/v maltodextrin solution (Myopure, Petersham, NSW, Australia). Liquid reinforcers were flavoured with either 0.4% v/v concentrated lemon juice (Berri, Melbourne, Victoria, Australia) or 0.2% v/v peppermint extract (Queen Fine Foods, Alderley, QLD, Australia) to provide unique sensory properties to each reinforcer. Liquids were delivered over a period of 0.33 s via a peristaltic pump corresponding to a volume of 0.2 mL. The volume and concentration of liquid reinforcers was chosen to match the calorific value of the corresponding grain pellet reward, and have been found to elicit similar rates of Pavlovian and instrumental responding as a pellet reward in other experiments conducted in this lab. In all experiments involving liquids, the magazine was scrubbed with warm water and thoroughly dried between sessions to remove residual traces of the liquid reinforcer. To reduce neophobia to the reinforcers, one day prior to magazine training sessions all animals were pre-exposed to the reinforcers (10 g of pellets per animal and 25 ml of liquid reinforcer per animal) in their home cage.

## Magazine training

I n all experiments, animals received two sessions of magazine training, one for each reinforcer with the following parameters: reward delivery was on an RT60 s schedule for 16 rewards with the house light and fan kept on throughout the session. Sessions were separated by at least 2 hr.

## Experiment 1. instrumental devaluation by LiCl taste aversion

All animals received 2 separate sessions of training each day with the pellet and sucrose rewards, an instrumental lever training session (lever extended) and a magazine training (lever retracted) session with non-contingent reward delivery to provide equivalent exposure to the alternative reward. The order of training sessions and the identity of the instrumental and alternate reward were fully counterbalanced across all groups. All training session were separated by a period of at least 2 hr.

First, animals were familiarised with lever training using a fixed ratio 1 schedule (FR1, reward delivered on each lever press), for 60 mins or until a maximum of 25 rewards were earned. The alternative, non-instrumental, reward was delivered on an RT30s (random time 30 s) schedule for 1 hr or until 25 rewards had been delivered.

Instrumental acquisition training occurred on the following 3 days. Instrumental training sessions lasted until 40 rewards were achieved and lever pressing was rewarded on a RI30s schedule (random interval 30 s such that on average every 30 s a reward becomes available to reward the next lever press). The alternate reward session involved an RT30s schedule for 40 rewards. The use of interval and time based schedules of reinforcement was designed to match the instrumental and alternate reward sessions so that all experiences were identical except for the presence (and response requirement) of the lever in the instrumental session.

Following devaluation of the reward by taste aversion, all animals were tested with the instrumental lever to assess devaluation. The test was conducted under extinction and the lever was extended for 10 mins. On the following day, all animals were given a 20 min re-acquisition test to assess devaluation in the presence of the instrumental reinforcer (RI30s schedule).

## Taste aversion

Following instrumental training all animals received taste aversion training on one of the reinforcers. Half the animals in each surgery condition (sham and lesion) were allocated to a devalued or a non-devalued group after being matched on their level of instrumental performance. The devalued groups received pairings of the instrumental reinforcer paired with 0.15M LiCl injections i.p. (15 mL/Kg) after 30 mins of individual access to that reinforcer in the devaluation chamber. The non-devalued groups received similar LiCl injections following access to the alternate (i.e. non-instrumental) reinforcer. On alternating days (order counterbalanced) all groups received 0.9% w/v saline injections (15 mL/Kg) following 30 mins access to the reinforcer that was not paired with LiCl. This procedure was repeated over 6 days such that all animals received 3 reinforcer-LiCl pairings and 3 reinforcer-saline pairings. Therefore, all animals had one reinforcer devalued with LiCl such that in the devalued groups it was the instrumental reinforcer and in the non-devalued groups it was the alternate reinforcer.

All animals were given an additional day without injections at the end of the taste aversion procedure before any further behavioural testing was conducted. This minimised the possibility of nausea persisting at test after the final LiCl injection, and ensured that all animals were at a comparable level of hunger at test.

## Experiment 2: pavlovian devaluation by LiCl taste aversion

### Histology and lesion group allocation

Lesion damage is depicted in *Figure 2A*. Lesion extent was judged by a trained observer blind to group allocation. Once approximate lesion extent was drawn, a second trained observer (also blind to surgical conditions) independently verified the extent of the drawn lesions and the grounds for exclusion. Animals were excluded if there was only unilateral OFC damage, evidence of damage to the dorsal part of the anterior olfactory nucleus ventral to OFC or if there was extensive damage to the white matter of the forceps minor of the corpus callosum. Seven animals were excluded due to the presence of infection that was evident across the entirety of the frontal cortex, and a further two animals were excluded due to illness throughout behavioural training. Three animals were excluded due to insufficient bilateral damage to OFC structures. Seven animals were excluded based on significant unilateral or bilateral damage to the dorsal part of the anterior olfactory nucleus. One animal was excluded due to almost complete unilateral damage to primary and secondary motor areas M1 and M2, ventral to the OFC. Final group numbers were sham $n = 13$, lesion $n = 31$ ($N = 44$).

The lesion drawings were then analysed to establish the extent of damage to the subregions of the OFC from which two distinct lesion groups could be formed. OFC lesions were predominantly confined to LO and DLO as in previous experiments and were distributed across a large anterior-posterior range. This observation was quantified by estimating the percentage of bilateral damage across all OFC structures at 7 coronal planes (+5.20 to +2.20 mm from bregma in steps of 0.50 mm). At each coronal plane the total area of each orbital structure and the total area of lesion damage were estimated (number of pixels counted using Adobe Photoshop CS; San Jose, CA). Bilateral damage was defined by comparing the hemisphere with the smallest lesion area for each orbital subregion and the total area of the structure in that hemisphere. Total OFC damage at each section was defined by the sum of damaged area relative to the sum of the total area of each orbital structure that is % Bilateral OFC damage $= 100 \ x \ \frac{Total \ lesion \ area}{Total \ orbital \ structure \ area}$. The OFC structures included in this analysis were LO, DLO, VO, AI, AId and AIv, however the damage (*Figure 2A*) was relatively confined to LO and DLO. Most animals had OFC damage at +3.70 mm from bregma, so anterior and posterior lesions were based on comparing relative lesion volume anterior (+5.20, +4.70, +4.20 mm) and posterior (+3.20, +2.70, +2.20) to this point. Animals with a greater lesion damage anterior to +3.70 were assigned to the anterior OFC lesion group, and animals with greater lesion damage posterior to +3.70 were allocated to the posterior OFC group. While these criteria for anterior and posterior OFC were based on the present sample, these criteria also define the anterior-posterior split that defines DLO and AId/AIv, and the presence of the forceps minor of the corpus callosum, which supports the external validity of these criteria.

Final lesion group numbers were anterior $n = 16$, posterior $n = 15$. A Group (anterior, posterior) x Plane (+5.20, +4.70, +4.20, +3.70, +3.20, +2.70, +2.20) mixed ANOVA analysing the percentage bilateral lesion volume (*Figure 2B*) revealed no significant overall effect of Group ($F_{(1, \ 29)}$=0.21,

p=0.65) but a significant main effect of Plane ($F_{(6, 174)}$=64.07, p<0.001) and Group x Plane interaction ($F_{(6, 174)}$=17.70, p<0.001). Follow up planned contrasts comparing groups at each coronal plane revealed greater damage in the anterior group at +4.70 ($F_{(1, 29)}$=7.87, p=0.01) and +4.20 ($F_{(1, 29)}$=33.74, p<0.001), and greater damage in the posterior group at +3.70 ($F_{(1, 29)}$=6.97, p=0.01) and +3.20 ($F_{(1, 29)}$=12.78, p=0.001) but no significant differences at +5.20 ($F_{(1, 29)}$=2.89, p=0.10) or +2.20 ($F_{(1, 29)}$=1.86, p=0.18) (*Figure 2B*). These differences indicate that the grouping criteria were effective at creating partially overlapping but distinct lesion groups.

## Acquisition

Pavlovian acquisition training occurred over 12 days involving one session of training per day. Each session consisted of 32 trials with a 90 s ITI, 15 s CS duration co-terminating with the delivery of a single reward. Two CS (5 Hz click and 78 dB white noise) and US (grain pellet and lemon sucrose) relationships were maintained throughout training such that rats always experienced 16 of each unique CS-US pairings each session (counterbalanced).

## Taste aversion

Taste aversion to one of the rewards (counterbalanced) was achieved by pairing reward consumption with nausea induced by Lithium Chloride (LiCl; Sigma-Aldrich, Switzerland). All rats received 3 pairings of one reward with an i.p. injection of 0.15 M LiCl (15 mL/Kg) and 3 pairings of the other reward with saline (0.9% w/v; Sigma-Aldrich, Switzerland). The first 2 food-injection pairings occurred immediately after providing rats with 30 mins free access to the reward in the devaluation chambers. The final food-injection pairings occurred in the test chamber after rats were exposed to a magazine training session with one of the reinforcers (reward delivered randomly on an RT60 s schedule for 16 rewards). The order of food-injection pairings was counterbalanced and alternated across the 6 days of taste aversion training. The final food-injection pairings in the test chamber were conducted to ensure that the taste aversion transferred between the devaluation chambers and the testing chambers. All animals were given an additional day without injections at the end of the taste aversion procedure before any further behavioural testing was conducted. This minimised the possibility of nausea persisting at test after the final LiCl injection, and ensured that all animals were at a comparable level of hunger at test.

## Devaluation test

Devaluation testing was identical to Pavlovian acquisition training except that it was performed under extinction that is no rewards were delivered throughout the session. The identity of the first and second cue and was predetermined at test to allow for counterbalancing. Animals were tested again on the following day with the identity of the first cue changed to fully counterbalance the test procedure.

## US specific reinstatement

After the final devaluation test, all rats received a US-specific reinstatement test to verify whether any failure of devaluation was due to impaired retention of the acquired taste aversion. On each day animals were pre-exposed to a single US type within the test chamber before being tested with 8 presentations in extinction of the CS that predicted the US. Exposure sessions involved a 5 min baseline period in which nothing happened in the chamber, followed by a reward delivery every 5 for 30 s (6 rewards), and then a post reward period of 5 mins. After the session, rats were temporarily returned to their home cage to allow for any remaining rewards to be collected for counting later and thorough cleaning of the reward site. Rats were then returned to the testing chamber for a test consisting of 8 CS presentations (90 s ITI) in extinction with the CS that predicted the recently delivered US. The order of outcome testing across both days was fully counterbalanced.

## Re-acquisition

After reinstatement testing, all rats received 3 days of re-acquisition training. These were identical to Pavlovian acquisition training except that only the CS paired with the non-devalued CS was presented for all 32 trials.

## Autoshaping

Following the reacquisition training, all animals were trained for 12 days on a discriminated autoshaping procedure where the non-devalued reward continued to serve as the US. Each session consisted of 32 trials with a 90 s ITI and 15 s CS duration, 16 rewarded CS+ trials and 16 non-rewarded CS- trials. The CS+ and CS- involved the insertion of the lever on the left or right hand side of the magazine (counterbalanced). Responding on the lever had no programmed consequences but was recorded for analysis.

## Reversal

Autoshaping was followed by reversal training for 12 days such that the CS+ and CS- contingencies were reversed that is the rewarded lever cue no longer predicted reward and the non-rewarded lever cue predicted reward.

## Locomotor screening

All animals were tested for locomotor activity before surgery, and again at the end of training, to verify the absence of any effects on locomotor activity in a within-subjects design.

## Statistical analysis

Baseline responding. Baseline rates of responding across all experiments did not differ between groups. Separate mixed ANOVAs on baseline responding in each experimental stage did not reveal significant main effects or interactions with Group (all $F < 1.75$, $p > 0.14$).

CS responding was operationalized as the time spent exploring the magazine during the 15 s CS period. PreCS responding was operationalized as the duration of responding during the 15 s immediately preceding the 15 s CS and was used as a measure of baseline responding to the testing context. All data were analysed with mixed ANOVAs, and significant interactions of interest were followed up with ANOVAs on the relevant subset of data. Following significant omnibus ANOVA tests, planned linear and quadratic orthogonal trend contrasts and their interactions between groups were analysed to assess differences in rates of responding.

## Acknowledgements

We gratefully acknowledge Fred Westbrook, Nathan Holmes, David Bannerman, and Mark Walton for their invaluable feedback. Research supported by grants awarded to Simon Killcross from the Australian Research Council (ARC Discovery Grant DP0989027 and DP120103564).

## Additional information

### Funding

| Funder | Grant reference number | Author |
| --- | --- | --- |
| Australian Research Council | DP0989027 | Simon Killcross |
| Australian Research Council | DP120103564 | Simon Killcross |

The funders had no role in study design, data collection and interpretation, or the decision to submit the work for publication.

### Author contributions

Marios C Panayi, Conceptualization, Formal analysis, Investigation, Methodology, Writing—original draft, Writing—review and editing; Simon Killcross, Conceptualization, Resources, Supervision, Funding acquisition, Methodology, Writing—review and editing

### Author ORCIDs

Marios C Panayi http://orcid.org/0000-0003-2635-5638

## Ethics

Animal experimentation: All animal research was carried out in accordance with the National Institute of Health Guide for the Care and Use of Laboratories Animals (NIH publications No. 80-23, revised 1996) and approved by the University of New South Wales Animal Care and Ethics Committee.

## Decision letter and Author response

Decision letter https://doi.org/10.7554/eLife.37357.013
Author response https://doi.org/10.7554/eLife.37357.014

## Additional files

### Data availability

All experimental data are publicly available at Dryad Digital Repository (DOI: 10.5061/dryad. c3b0260)

The following dataset was generated:

| Author(s) | Year | Dataset title | Dataset URL | Database, license, and accessibility information |
|---|---|---|---|---|
| Panayi MC | 2018 | Data from: Functional heterogeneity within the rodent lateral orbitofrontal cortex dissociates outcome devaluation and reversal learning deficits | http://dx.doi.org/10.5061/dryad.c3b0260 | Available at Dryad Digital Repository under a CC0 Public Domain Dedication |

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

# Appendix 1

DOI: https://doi.org/10.7554/eLife.37357.008

## Effect of OFC lesions on sensory specific pavlovian to instrumental transfer

Similar to *Ostlund and Balleine, (2007a)*, we tested whether the effects of our specific OFC lesions affected the use of sensory-specific Pavlovian information to guide instrumental responding using a Pavlovian to instrumental transfer test (PIT).

## Methods

### Acquisition training

On each day all animals received either a single Pavlovian training session, or two instrumental training sessions. The order of Pavlovian and instrumental sessions alternated each day.

### Pavlovian training

All animals received a total of 16 days of Pavlovian training. Pavlovian training sessions consisted of 3 CSs, a 2800 Hz, 80 dB tone, 78 dB white noise and a 5 Hz train of clicks. There were 4 presentations of each cue (i.e. a total of 12 cues presented within a session) each lasting 2 mins with a variable ITI of 300s. Reward was delivered throughout the cue period on a RT 30s schedule. Each cue was paired with a unique outcome (grain pellet, lemon sucrose, and peppermint maltodextrin) and the identity of that outcome remained constant. All unique cue-outcome combinations were counterbalanced across animals and within groups.

### Instrumental training

Prior to Pavlovian and instrumental acquisition training all animals were given 2 days of lever training on a continuous reinforcement schedule (each lever press was rewarded) using the same parameters as the instrumental training sessions.

All animals received a total of 12 days of instrumental training. Instrumental training involved two sessions per day, separated by at least one hour. During the session a single lever was extended and lever pressing was rewarded with a unique liquid outcome, either lemon sucrose or peppermint maltodextrin. During the second instrumental session of the day, a different lever was extended and lever pressing was rewarded with the unique liquid outcome that was not paired with the earlier lever. The identity of the lever outcome pairings was kept constant throughout training and was counterbalanced between subjects and within groups. After initial lever acquisition, animals received three days of Random interval RI 15s, three days of RI 30s and six days of RI 60s.

### PIT test

The PIT test involved a single lever presented at the start of the session for 10 mins with no programmed consequences to extinguish lever pressing behavior to a low baseline rate (this allows for clearer demonstration of the potential rate-enhancing effect of CS presentations). Then the CSs were played for 2 min with a fixed 2 min inter-stimulus interval. Each CS was played three times (a total of 9 CS presentations) and the order of CS presentation was randomized. Throughout the session no rewards were delivered and lever pressing and magazine entry were recorded with no programmed consequences. A second identical test session was conducted on the following day using the lever that had yet to be tested. Order of lever presentation was counterbalanced. This pattern of tests was repeated once after 4 days of retraining on Pavlovian and instrumental sessions.

## Results

### Histology

Lesion damage is depicted in (**Appendix 1—figure 1A** ). One sham animal was excluded due to extensive damage to primary and secondary motor areas M1 and M2. Final $N$ = 15; sham $n$ = 7, lesion $n$ = 8.

### Behavioural results

#### Pavlovian

A mixed Group x Day (16 days) x US (sucrose, maltodextrin, pellet) ANOVA was conducted on CS-PreCS magazine entry rate to quantify Pavlovian acquisition. This analysis revealed that responding was greater for pellets than sucrose reinforcers ($F_{(1, 13)}$=8.69, p=0.03, main effect of US, $F_{(2, 26)}$=4.56, p=0.02, no other differences between reinforcers reached significance, sucrose vs maltodextrin, $F_{(1, 13)}$=0.37, p=0.91, maltodextrin vs pellets, $F_{(1, 13)}$=4.42, p=0.16). Surprisingly, CS responding was significantly greater in the sham than the lesion group ($F_{(1, 13)}$=12.49, p=0.004). Importantly, there were no significant interactions between Group, Day, or US (remaining $F$ < 1.35, p>0.11). Both groups showed significant acquisition over days of training (significant main effect of Day, $F_{(15, 195)}$=4.86, p<0.001, significant positive linear, $F_{(1, 13)}$=8.69, p<0.001, and negative quadratic trend, $F_{(1, 13)}$=6.29, p=0.03), and responding (entries per minute, CS-PreCS) on the final day of acquisition were sham ($M$ = 7.76, $SD$ = 1.36) lesion ($M$ = 6.03, $SD$ = 1.27). It is likely that the increased responding to the pellet reinforcer is a result of the use of magazine frequency as a measure as we routinely observe the opposite pattern when using magazine duration as a measure (e.g. Experiment 2). Unfortunately, magazine duration data were not recorded during this experiment to determine whether the difference between sham and lesion groups was only present on this measure. Furthermore, it is important to note that this measure of Pavlovian conditioning is conflated with consummatory responses since the USs were delivered at random times throughout the CS, and as such it is hard to draw clear conclusions about any observed differences in responding.

Instrumental acquisition did not differ between groups, a pattern supported by a mixed Group x Day (12 days) ANOVA finding a significant main effect of Day ($F_{(11, 143)}$=58.15, p<0.001) but no significant effect of Group ($F_{(1, 13)}$=1.62, p=0.23) or Group x Day interaction ($F_{(11, 143)}$=1.10, p=0.36). Response levels (lever presses per minute) on the final day of instrumental training were similar in sham ($M$ = 11.00, $SD$ = 3.66) and lesion ($M$ = 9.12, $SD$ = 3.61) groups.

Extinction of magazine and lever responding in the 10 min prior to testing did not reveal any group differences (Figure S1 B, C). Separate mixed Group x Block (10 blocks of 1 min) ANOVAs on lever pressing and magazine approach revealed significant main effects of Block (lever pressing $F_{(9, 117)}$=8.95, p<0.001, magazine entries $F_{(9, 117)}$=2.60, p=0.01) but no effect of Group or Group x Block interactions (remaining $F$ < 1.42, p>0.19).

At test, lever pressing was assessed in the presence of the CSs that either predicted the same outcome as the instrumental response, a different outcome (predicted by the alternative instrumental response) or a general outcome not predicted by either instrumental response. In both groups lever pressing was potentiated most by CS same, moderately by CS different and minimally by CS general (Figure S1 D). A mixed Group x Cue (same, different, general) ANOVA confirmed that responding differed between cues (main effect of Cue $F_{(2, 26)}$=6.32, p=0.01) but was not differentially affected by lesion group (main effect of Group $F_{(1, 13)}$=0.04, p=0.85, Group x Cue interaction $F_{(2, 26)}$=1.26, p=0.30). Bonferroni adjusted simple main effects revealed that responding to CS same was greater than CS general ($F_{(1, 13)}$=10.25, p=0.02), however CS same did not differ from CS different ($F_{(1, 13)}$=3.58, p=0.24) and CS different did not differ from CS general ($F_{(1, 13)}$=3.92, p=0.21).

Additional comparisons examined whether responding to each cue was significantly different from baseline (i.e. 0). The data were collapsed across groups as there was no significant interaction with group. Lever responding was significantly greater than baseline for

CS Different ($F_{(1, 13)}$=8.29, p=0.01) and CS Same ($F_{(1, 13)}$=20.20, p=0.001), but not for CS General ($F_{(1, 13)}$=0.35, p=0.56) which suggests that there was no significant evidence of a general PIT effect for CS General.

Magazine responding during the test session was not differentially affected by either group or cues (Figure S1 E). A mixed Group x Cue (same, different, general) ANOVA supported this observation with all effects failing to reach significance (all $F$ < 1.23, p>0.29).

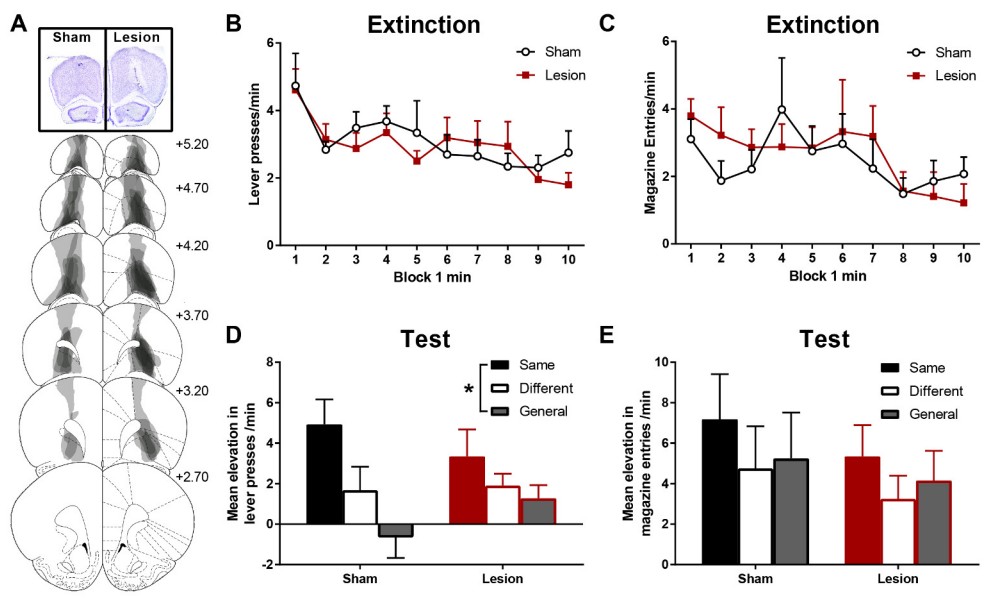

**Appendix 1—figure 1.** The effects of excitotoxic OFC lesions on specific Pavlovian-to-instrumental transfer (PIT). (**A**) Representative OFC lesion damage in the lesion group. Semi-transparent grey patches represent lesion damage in a single subject, and darker areas represent overlapping damage across multiple subjects. Coronal sections are identified in mm relative to bregma (Paxinos and Watson, 1997). Rate of lever pressing (**B**) and magazine entry behaviour (**C**) during extinction of the instrumental response prior to PIT testing. Instrumental lever pressing (**D**) and magazine entry behaviour during the specific PIT test. Responding plotted as the mean response rate per minute during each cue minus the preceding baseline no-cue period. Same and different conditions indicate whether the Pavlovian CS predicted the same or different liquid reinforcer to the instrumental response, the general condition indicates responding during the CS that predicted pellets which were never an instrumental reinforcer. Error bars depict + SEM. (*) Symbol denotes statistical significance of simple or main effects following a significant interaction.
DOI: https://doi.org/10.7554/eLife.37357.009

