## [Decision Letter]

Thank you for sending your article entitled "The rodent lateral orbitofrontal cortex represents expected Pavlovian outcome value but not identity" for peer review at *eLife*. Your article is being evaluated by three peer reviewers, and overseen by Geoffrey Schoenbaum as the Reviewing Editor and Sabine Kastner as the Senior Editor.

Given the list of essential revisions, including new experiments, the editors and reviewers invite you to respond within the next two weeks with an action plan and timetable for the completion of the additional work. We plan to share your responses with the reviewers and then issue a binding recommendation.

Overall the reviewers felt that the experiments addressed an extremely interesting and important set of questions, yielding intriguing and valuable results. However, all three reviewers had major concerns about the core experiment in Figure 2, claimed to show that the orbitofrontal cortex is not necessary for devaluation by satiety and, by extension, the authors strong conclusions that the orbitofrontal cortex is therefore not representing information about outcome identity. The reviews express the variety of these concerns, but perhaps the most critical is that the negative effect there comes against a backdrop of a very weak positive effect in controls. So, after discussions, it was felt that even with appropriate caveating and consideration of alternatives, it was not possible to accept this experiment. Two solutions were discussed.

1) Remove this experiment and focus the paper on the important and novel findings in Figure 4, Figure 5 and Figure 6. This would obviously mean extensive changes to the framing of the paper and the claims based on the negative results in Figure 2. However, this is not new experimental work. And it would allow the publication of what all the reviewers agreed were well-done and important and novel results.

2) Repeat experiment 2, ideally using a within subjects design where each animal is tested under both forms of devaluation or in a way that differences in lesion location are not an issue, and with rewards that differ only in sensory and not motivational properties. This would represent substantially more work, but would directly test the question at issue, without confounds of lesion or others noted. If this were done, it would obviously address the concerns and would be a great experiment, assuming the effects (positive or negative) were statistically robust and clear.

The reviewers felt that it was worth hearing a response from the authors before making a final decision.

Reviewer #1:

In this study the authors aim to clarify the role of central/lateral orbitofrontal cortex in a variety of different behaviors – instrumental versus Pavlovian behavior, illness versus satiety induced devaluation, and also reversal learning and sign tracking. In the process, they show that neurotoxic lesions of rat orbitofrontal cortex do not affect instrumental devaluation (between subjects) or Pavlovian devaluation by specific satiety (within subjects). They confirm that orbital lesions do affect Pavlovian devaluation by illness (within subjects), whether lesions were "anterior" or "posterior", and they extend this finding by showing that presentation of the outcomes prior to testing restores normal behavior (somewhat like satiety). In addition to these results, they show that specific transfer is sensitive to satiety effects. And they produce novel data suggesting that their more posterior lesions also cause deficits in allocation of behavior in a sign tracking task, both during acquisition and reversal learning. From these results, the authors draw fairly strong conclusions regarding this collection of areas. Overall, I think the areas of conflicts the authors have tried to address are important, and the data they provide are valuable. They represent a wealth of diverse and very interesting results. However, the designs and how the studies fit together are not ideal, and as a result I am uncomfortable with the strong conclusion and framework the authors put on their data. I think their results with regard to Pavlovian devaluation are open to a variety of other interpretations that are basically not acknowledged. I also think that the paper would be simpler and clearer if it were divided into two. That is, the points regarding devaluation are dissociable from the points regarding reversal learning and sign tracking.

Regarding the devaluation experiments, the main conclusion is that the orbitofrontal cortex is not involved in representation of information about the sensory features of expected outcomes. This is based on two pieces of information: (1) the finding that pre-training orbitofrontal lesions did not affect Pavlovian devaluation by selective satiation, whereas similar lesions do affect Pavlovian devaluation by illness (here and elsewhere) and (2) the idea that satiety, unlike illness, can affect responding either through a change in the desirability of the outcome or via a sort of habituation effect on the sensory representation of the outcome. The authors seem to be proposing that the orbitofrontal cortex (writ large) is doing the former but not the latter. I have several problems with basing this strong conclusion on these data.

First, I think that to the extent there is a second way that satiety (because of its proximity to testing) but not illness (because of its distance) can affect behavior, the results may show that the area lesioned is not required for this sensory habituation mechanism, but they cannot show that the area is not involved in the representation of the information. And it is clear from numerous studies that value-neutral information about predicted events is represented in the areas targeted here, in rats, monkeys and humans. Given this, the strong conclusion, expressed even in the title, seems unwarranted.

Second, I think the authors are ignoring causal studies that suggest the information about upcoming sensory events in orbitofrontal cortex is necessary for behavior, at least in some circumstances. For example, inactivation of this area affects performance and learning that depends on such information in a sensory preconditioning task (Jones et al., 2012). Also, while pre-training lesions do not affect specific transfer, post-training lesions or post-training manipulations of afferents from amygdala do affect specific transfer (Lichtenberg et al., 2018; Ostlund and Balleine, 2007). The authors do not discuss these results, but I think they are quite important. At a minimum they show that if the brain is allowed to learn normally, the orbitofrontal cortex is in fact necessary for using information about identity. Perhaps the intact performance here reflects how the lesioned brain learns the original information, rather than some dichotomy in how information about reward identity is used in devaluation by illness versus satiety.

In any event, in light of these other studies, I think the failure to affect satiety here ends up looking more like a special case or an exception to the rule than a strong demonstration that orbitofrontal cortex either does not represent this information or has it but is not using it.

In addition, I think there are other differences between illness and satiety that are relevant and not considered. One is simply the strength of the effect. Illness is presumably stronger than satiety in changing behavior, both generally and in the two studies that are compared here. Perhaps it is this weakness that supports the negative finding and not a fundamental difference in the areas involvement in devaluation by satiety. Another difference is that satiety is something that can be experienced previously during training, since over the course of training the rats gain experience with the cues in various stages of satiation. This prior experience could allow for the adjustment of behavior to some extent without the sort of processing proposed to depend on orbitofrontal cortex. Illness induced devaluation is not subject to this potential confound. Third it seems to me that the two outcomes used here differ not only in their sensory properties but also in their motivational basis. Pellets satisfy hunger, whereas sucrose satisfies hunger and thirst. Possibly the selective satiation affects behavior through these motivational mechanisms and that is what is not dependent on orbitofrontal cortex? Any or some combination of these proposals could explain the negative effects.

Finally, I think there are also differences in the lesions in the satiety and devaluation experiments – at least it looks to me like the lesions in figure 4 are larger and more lateral, which I think would be more likely to yield effects. Hindsight is easy, but a within subjects design for this critical comparison would have been better. As it stands, there are key lateral areas that seem to be more involved in the experiment where there are lesion effects.

Despite these issues, I do regard the individual experiments as well done and the overall topic is excellent and worth investigating, so I think the authors have valuable results that can help inform future work. I could imagine a different manuscript that more fairly considers these and perhaps other alternatives.

My preference would also be to separate the devaluation studies from those on reversal and sign tracking data. I fewer issues with these data and their interpretation, maybe because their conclusions are less strong and more alternatives are considered. But I think their presence detracts from a full consideration of the other data.

Reviewer #2:

This manuscript presents data from several experiments in rats that focus on the effects of orbitofrontal cortex lesions on devaluation. Individual experiments vary in the type of devaluation (specific satiety, taste aversion), lesion location (anterior, posterior), or learning (Pavlovian, instrumental). The main conclusion is that OFC does not represent sensory features of the outcome. This is based primarily on the finding that OFC lesions only diminish Pavlovian devaluation by taste aversion but not specific satiety (in conjunction with the assumption that habituation, leading to an inability to represent outcome features, is the defining difference between the two devaluation methods). In addition, there is evidence regarding a functional differentiation between anterior and posterior OFC.

This manuscript is well written and presents a large amount of interesting experimental data. My overall concern is that the conclusions are not well justified. They are based on a combination of null results and qualitative comparisons between different experiments which differ in more than one factor. For instance, the two key experiments (Figure 2 and Figure 4) differ not only in devaluation method but also in lesion location (and there is evidence that this may be important). In addition, besides habituation, there are other differences between the two devaluation methods that may explain the current findings and considering these would change the conclusions.

1) The main conclusion that OFC does not represent specific outcome properties is based on an assumption about the key difference between devaluation by specific satiety and taste aversion. Because specific satiety involves repeated exposure to the US, it "involves habituation of the sensory systems required to represent the sensory properties of the outcome." There are two issues with this assumption. (1) Even if specific satiety induces habituation of the sensory systems, it is unclear whether this would also diminish representations of predicted sensory features in OFC and the ability to retrieve the updated value of these outcomes. (2) There are more prominent differences between the two forms of devaluation that may explain the current results. Most importantly, whereas specific satiety slightly reduces the value of the US, taste aversion involves a much more dramatic change in value and can thus be expected to result in a relatively stronger devaluation effect. Indeed, comparing the effects of the two devaluation methods on behavior in the Sham group shows that taste aversion reduces magazine activity by about 40% (Figure 4F), whereas specific satiety changes behavior by only about 20% (Figure 2C). Thus, differences between the two methods could simply result from differences in their effectiveness. If the effect of devaluation by specific satiety is too small in the Sham group, the probability to detect differences between groups decreases.

2) In line with this, effects of Pavlovian devaluation by satiety were modest (Figure 2C). The authors report a significant main effect of devaluation, but are the effects within each group (Sham and lesion) individually significant? The devaluation effect in the OFC group appears smaller compared to the Sham group. Is it possible that the devaluation effect is driven by the Sham but not the Lesion group, and that the experiment was underpowered to find group differences?

3) The conclusion that OFC lesions affect Pavlovian devaluation by taste aversion, but not specific satiety is based on a positive and a null result from different experiments. However, the lesions for Pavlovian devaluation by satiety were more anterior compared to the lesions in the taste aversion experiment (Figure 2A and Figure 4A). Thus, the two experiments differ not only in the devaluation method but also in lesion location. Given that the results of Pavlovian devaluation by taste aversion may depend on where the lesion is made in OFC, this difference should not be taken lightly. To convincingly demonstrate that the devaluation method matters, both methods should be compared directly within the same experiment and with the same lesion location.

Reviewer #3:

In this report, the effects of lateral OFC lesions on a variety of behavioral tasks are presented. First, replicating previous results, pre-training LO lesions are shown not to affect sensitivity of instrumental responding to devaluation, suggesting intact action-outcome signaling. In contrast to previous reports, LO lesions are shown not to affect sensitivity of a Pavlovian CR to devaluation induced by sensory-specific satiety. The expression of specific PIT is shown to be sensitive to sensory-specific satiety, which the authors interpret as an indication that sensory-specific satiety acts via habituation of the sensory-specific features of the outcome and, thus, that LO is not required for representing such features. The most interesting and clear finding is the dissociability of anterior v. posterior OFC necessity for sensitivity of the Pavlovian response to devaluation, sign-tracking, and reversal learning. This report has many strengths including the thorough statistical reporting, multi-faceted behavioral analysis, combination of behaviors, and anterior v. posterior OFC analysis. The anterior v. posterior OFC lesions effects are especially exciting. Enthusiasm is somewhat diminished by the lack of clear support for many of the main conclusions (described below) and the limitations of the lesion approach used here. The relevance of this data to the broad readership of *eLife* over a more specialized journal is also not immediately clear, though this could be remedied.

1) My primary concern is that many of the conclusions in this report do not appear to be fully supported.

a) "[…] directly confirm the dissociable role of the rodent OFC in Pavlovian but not instrumental behavioural flexibility following outcome devaluation". While I don't totally disagree, this is not directly shown here because there is no direct comparison between instrumental and Pavlovian CR sensitivity to the same form of devaluation within subjects with the same lesions. Indeed, the extent of the lesion, esp. in the lateral and posterior domains was different between the data in Figure 1 and Figure 2 and Figure 4. I suggest removing or tempering this language.

b) "OFC lesions in rodents only disrupt the Pavlovian outcome devaluation effect when outcome value is manipulated by taste aversion but not specific satiety." This cannot be interpreted here for two reasons. First, this was not directly assessed between subjects with similar lesions. Indeed, the extent of the lesion is quite different for the subjects in Figure 2 v. Figure 4 and that is a major important finding that more posterior but not anterior lesions disrupt sensitivity of the Pavlovian CR to devaluation, leaving open the possibility that more posterior OFC lesions would also disrupt sensitivity to devaluation by sensory-specific satiety. Second, the efficacy of the sensory-specific satiety manipulation was not confirmed here with post-test consumption. Indeed, the sensory-specific satiety devaluation effect is not very convincing here in control subjects.

c) "Using a specific PIT test, we establish that, unlike taste aversion devaluation, specific satiety devaluation can act via a reduction in the efficacy of sensory specific outcome properties". While it is intriguing that PIT is sensitivity to sensory-specific satiety devaluation, to support this interpretation the authors would have to show that with their specific PIT procedures that PIT is not sensitive to taste aversion devaluation. It remains plausible that these findings indicate that during PIT subjects are using the CS to retrieve a representation of the outcome and its current value then determines whether the rat will press or not. Moreover, that these graphs lack any indication of error or variance makes these data difficult to interpret and, on top of this, the PIT effect size at ~1 press/min is much smaller than previous reports, including most of the reports cited as previous evidence for lack of PIT sensitivity to taste aversion, likely owing to the different procedures used.

d) The title: "The rodent lateral orbitofrontal cortex represents expected Pavlovian outcome value but not identity". Loss of function studies cannot tell you what information is represented or especially what information is not represented, this would require recordings. Moreover, that all the lesions were pre-training further confounds ability to interpret their effects on initial learning, updating, v. retrieval of stimulus-outcome memories. I suggest changing the title to better reflect the type of assessment provided by the data.

2) The difference in taste aversion and specific satiety results here is hypothesized to result from the possibility of sensory-specific satiety acts via habituation of the sensory-specific features of the outcome. This becomes crucial for the interpretation of the lesion results. But there are other differences between taste aversion and sensory-specific satiety. Taste aversion is certainly more severe; it is also a permanent, consolidated memory. This possibility and its relevance to interpretation of the OFC lesions results should be considered.

3) There are several previous findings that are seemingly contradictory to these results that are not, but should be discussed here, including data showing the representation of reward features and identity in human lateral OFC (see recent work from Kahnt and O'Doherty labs), evidence that post-training OFC lesions disrupt expression of specific PIT (Ostlund and Balleine), and evidence that inactivation of amygdala projections to lateral OFC disrupts both specific PIT and sensitivity of the Pavlovian CR to sensory-specific satiety devaluation (recent Wassum lab paper). These recording papers are also of relevance to the interpretation here: Mcdannald et al., 2014 Stalnaker et al., 2014.

4) The lesions here are purported to be "focal LO" lesions. By my eye they include VLO and this should be clarified.

[Editors' note: further revisions were requested prior to acceptance, as described below.]

Thank you for resubmitting your work entitled "Functional heterogeneity within the rodent lateral orbitofrontal cortex dissociates outcome devaluation and reversal learning deficits" for further consideration at *eLife*. Your revised article has been favorably evaluated by Sabine Kastner (Senior Editor), a Reviewing Editor, and three reviewers.

The manuscript has been improved but there are some remaining issues that need to be addressed before acceptance, as outlined below. Most important were issues #1 and 3 raised by reviewer 3.

Reviewer #3:

In this revised and more focused report the authors replicate previous findings that lOFC lesions disrupt sensitivity of Pavlovian conditional responding to devaluation, but do not disrupt instrumental devaluation. They nicely demonstrate the Pavlovian devaluation effect for both anterior and posterior lOFC lesions. They also show that posterior lOFC lesions disrupt acquisition of a sign-tracking conditional response and the reversal of this response, whereas anterior lOFC lesions spare this behavior. The report provides a nice addition to the OFC literature. I do have a few remaining concerns regarding the interpretation and discussion of the data.

1) The data in Figure 2G are interpreted as demonstrating that the anterior and posterior lOFC lesions did not cause "a failure to acquire sensory specific cue-outcome associations" and that the deficits in Pavlovian devaluation are "specific to recalling the new value of the devalued outcome and/or integrating it into appropriate behavioural control". See subsection “Outcome Devaluation”, subsection “Rodent and primate homology” and subsection “Theoretical accounts of OFC function”. I don't think this claim can be made for two reasons. (1) During this test the animals get feedback about their entries that will change the likelihood that they will enter subsequently. Thus, they need not have learned the specific cue-outcome relationships to show reduced performance during the devalued CS in this task. (2) The lesion was not restricted to training or test, thus making it impossible to distinguish learning v. recall effects- of which both are likely at play. Thus, it cannot be concluded that the lOFC is selectively important for recalling the new value or outcome information, or with a "selective role for the OFC in Pavlovian model-based inferences". To be clear, I do not disagree that the lOFC is important for retrieving outcome information, there are clear data that support its function in this regard, but the pre-training lesion data here cannot be used to support a selective role in retrieval or inference and not one in encoding.

2) It remains the case that the findings of deficits in initial acquisition from posterior lOFC lesions makes it difficult to interpret a reversal learning deficit here.

3) There is very little consideration of the limitations of the pre-training lesion approach selected here. Encoding v. retrieval, compensatory mechanisms, etc. These limitations are important to understanding the present results and should be discussed. Also, in my opinion the discussion is too long. It reads more like a review paper than a discussion of the present results. I suggest making it much shorter.

4) Figure 1D. I still think the devaluation effect is weak in the Sham group and worry these data might color the report for readers. Perhaps plotting the data over time might reveal a bigger effect earlier in the test.

---

## [Author Response]

[Editors' note: the authors’ plan for revisions was approved and the authors made a formal revised submission.]Reviewer #1:[…] Despite these issues, I do regard the individual experiments as well done and the overall topic is excellent and worth investigating, so I think the authors have valuable results that can help inform future work. I could imagine a different manuscript that more fairly considers these and perhaps other alternatives.My preference would also be to separate the devaluation studies from those on reversal and sign tracking data. I fewer issues with these data and their interpretation, maybe because their conclusions are less strong and more alternatives are considered. But I think their presence detracts from a full consideration of the other data.

These concerns are addressed by the removal of experiments 2, 3, and 5 from the manuscript. We thank the reviewer for this considered review and agree that the more focused manuscript allows for a more careful consideration of the remaining data.

Reviewer #2:1) The main conclusion that OFC does not represent specific outcome properties is based on an assumption about the key difference between devaluation by specific satiety and taste aversion. Because specific satiety involves repeated exposure to the US, it "involves habituation of the sensory systems required to represent the sensory properties of the outcome." There are two issues with this assumption. (1) Even if specific satiety induces habituation of the sensory systems, it is unclear whether this would also diminish representations of predicted sensory features in OFC and the ability to retrieve the updated value of these outcomes. (2) There are more prominent differences between the two forms of devaluation that may explain the current results. Most importantly, whereas specific satiety slightly reduces the value of the US, taste aversion involves a much more dramatic change in value and can thus be expected to result in a relatively stronger devaluation effect. Indeed, comparing the effects of the two devaluation methods on behavior in the Sham group shows that taste aversion reduces magazine activity by about 40% (Figure 4F), whereas specific satiety changes behavior by only about 20% (Figure 2C). Thus, differences between the two methods could simply result from differences in their effectiveness. If the effect of devaluation by specific satiety is too small in the Sham group, the probability to detect differences between groups decreases.

These concerns are addressed by the removal of experiments 2, 3, and 5 from the manuscript.

2) In line with this, effects of Pavlovian devaluation by satiety were modest (Figure 2C). The authors report a significant main effect of devaluation, but are the effects within each group (Sham and lesion) individually significant? The devaluation effect in the OFC group appears smaller compared to the Sham group. Is it possible that the devaluation effect is driven by the Sham but not the Lesion group, and that the experiment was underpowered to find group differences?

These concerns are addressed by the removal of experiments 2, 3, and 5 from the manuscript.

3) The conclusion that OFC lesions affect Pavlovian devaluation by taste aversion, but not specific satiety is based on a positive and a null result from different experiments. However, the lesions for Pavlovian devaluation by satiety were more anterior compared to the lesions in the taste aversion experiment (Figure 2A and Figure 4A). Thus, the two experiments differ not only in the devaluation method but also in lesion location. Given that the results of Pavlovian devaluation by taste aversion may depend on where the lesion is made in OFC, this difference should not be taken lightly. To convincingly demonstrate that the devaluation method matters, both methods should be compared directly within the same experiment and with the same lesion location.

These concerns are addressed by the removal of experiments 2, 3, and 5 from the manuscript.

Reviewer #3:[…] 1) My primary concern is that many of the conclusions in this report do not appear to be fully supported.a) "[…] directly confirm the dissociable role of the rodent OFC in Pavlovian but not instrumental behavioural flexibility following outcome devaluation". While I don't totally disagree, this is not directly shown here because there is no direct comparison between instrumental and Pavlovian CR sensitivity to the same form of devaluation within subjects with the same lesions. Indeed, the extent of the lesion, esp. in the lateral and posterior domains was different between the data in Figure 1 and Figure 2 and Figure 4. I suggest removing or tempering this language.

These concerns are addressed by the removal of experiments 2, 3, and 5 from the manuscript.

b) "OFC lesions in rodents only disrupt the Pavlovian outcome devaluation effect when outcome value is manipulated by taste aversion but not specific satiety." This cannot be interpreted here for two reasons. First, this was not directly assessed between subjects with similar lesions. Indeed, the extent of the lesion is quite different for the subjects in Figure 2 v. Figure 4 and that is a major important finding that more posterior but not anterior lesions disrupt sensitivity of the Pavlovian CR to devaluation, leaving open the possibility that more posterior OFC lesions would also disrupt sensitivity to devaluation by sensory-specific satiety. Second, the efficacy of the sensory-specific satiety manipulation was not confirmed here with post-test consumption. Indeed, the sensory-specific satiety devaluation effect is not very convincing here in control subjects.

These concerns are addressed by the removal of experiments 2, 3, and 5 from the manuscript.

c) "Using a specific PIT test, we establish that, unlike taste aversion devaluation, specific satiety devaluation can act via a reduction in the efficacy of sensory specific outcome properties". While it is intriguing that PIT is sensitivity to sensory-specific satiety devaluation, to support this interpretation the authors would have to show that with their specific PIT procedures that PIT is not sensitive to taste aversion devaluation. It remains plausible that these findings indicate that during PIT subjects are using the CS to retrieve a representation of the outcome and its current value then determines whether the rat will press or not. Moreover, that these graphs lack any indication of error or variance makes these data difficult to interpret and, on top of this, the PIT effect size at ~1 press/min is much smaller than previous reports, including most of the reports cited as previous evidence for lack of PIT sensitivity to taste aversion, likely owing to the different procedures used.

These concerns are addressed by the removal of experiments 2, 3, and 5 from the manuscript.

d) The title: "The rodent lateral orbitofrontal cortex represents expected Pavlovian outcome value but not identity". Loss of function studies cannot tell you what information is represented or especially what information is not represented, this would require recordings. Moreover, that all the lesions were pre-training further confounds ability to interpret their effects on initial learning, updating, v. retrieval of stimulus-outcome memories. I suggest changing the title to better reflect the type of assessment provided by the data.

These concerns are addressed by the removal of experiments 2, 3, and 5 from the manuscript.

2) The difference in taste aversion and specific satiety results here is hypothesized to result from the possibility of sensory-specific satiety acts via habituation of the sensory-specific features of the outcome. This becomes crucial for the interpretation of the lesion results. But there are other differences between taste aversion and sensory-specific satiety. Taste aversion is certainly more severe; it is also a permanent, consolidated memory. This possibility and its relevance to interpretation of the OFC lesions results should be considered.

These concerns are addressed by the removal of experiments 2, 3, and 5 from the manuscript.

3) There are several previous findings that are seemingly contradictory to these results that are not, but should be discussed here, including data showing the representation of reward features and identity in human lateral OFC (see recent work from Kahnt and O'Doherty labs), evidence that post-training OFC lesions disrupt expression of specific PIT (Ostlund and Balleine), and evidence that inactivation of amygdala projections to lateral OFC disrupts both specific PIT and sensitivity of the Pavlovian CR to sensory-specific satiety devaluation (recent Wassum lab paper). These recording papers are also of relevance to the interpretation here: Mcdannald et al., 2014 Stalnaker et al., 2014.

These concerns are addressed by the removal of experiments 2, 3, and 5 from the manuscript. Additional consideration of OFC heterogeneity in relation to the specific PIT effect are discussed in the Discussion section.

4) The lesions here are purported to be "focal LO" lesions. By my eye they include VLO and this should be clarified.

This is an important consideration given the claims of functional heterogeneity within LO. We note here that bilateral damage was mostly confined to LO. Bilateral damage to VO was minimal when present. We examined whether there were any specific correlations between our estimates of damage to any of the OFC subregions and devaluation/sign-tracking/reversal effects but found no significant relationships that might indicate that damage to any of these other regions might be important for these effects.

It is also of note that the designation of VLO is not always used. This region was originally demarcated as the orbital surface directly above the ‘orbital notch’ (Krettek and Price, 1977). This places the structure half-way between VO and LO (which we have distinguished as medial and lateral to the orbital notch). Close inspection of a number of lesion and neuroanatomical projection studies indicate that VLO may not be a unique structure, and most effects attributed to this structure are driven by damage to VO e.g. an attentional circuit appears to exist between VO, dorsocentral striatum, and posterior parietal cortex (Burcham et al., 1997; Cheatwood, Corwin and Reep, 2005; Cheatwood, Reep and Corwin, 2003; Conte et al., 2008; Corwin et al., 1994; Corwin and Reep, 1998; King, Corwin and Reep, 1989; Reep, Cheatwood and Corwin, 2003; Reep, Corwin and King, 1996; Reep et al., 1994; Reep and Corwin, 1999, 2009; Van Vleet et al., 2000; VanVleet et al., 2002; Vargo et al., 1988). However, we have not directly tested or dissociated the adjacent VO/LO regions, so we can only rely on secondary sources to support this conclusion. It is clear that our understanding of neuroanatomical and functional distinctions within the rodent OFC is still quite poor.

[Editors' note: further revisions were requested prior to acceptance, as described below.]Reviewer #3:[…] 1) The data in Figure 2G are interpreted as demonstrating that the anterior and posterior lOFC lesions did not cause "a failure to acquire sensory specific cue-outcome associations" and that the deficits in Pavlovian devaluation are "specific to recalling the new value of the devalued outcome and/or integrating it into appropriate behavioural control". See subsection “Outcome Devaluation”, subsection “Rodent and primate homology” and subsection “Theoretical accounts of OFC function”. I don't think this claim can be made for two reasons. (1) During this test the animals get feedback about their entries that will change the likelihood that they will enter subsequently. Thus, they need not have learned the specific cue-outcome relationships to show reduced performance during the devalued CS in this task. (2) The lesion was not restricted to training or test, thus making it impossible to distinguish learning v. recall effects- of which both are likely at play. Thus, it cannot be concluded that the lOFC is selectively important for recalling the new value or outcome information, or with a "selective role for the OFC in Pavlovian model-based inferences". To be clear, I do not disagree that the lOFC is important for retrieving outcome information, there are clear data that support its function in this regard, but the pre-training lesion data here cannot be used to support a selective role in retrieval or inference and not one in encoding.

This is a valid point. In this test the animals are briefly re-exposed to the USs, removal of the animal from the chamber, cleaning with hot water before drying the magazine, and returning to the chamber for the test. It is possible that the re-exposure alone has caused some aversion/withdrawal/inhibitory response to the magazine location/magazine approach action. This limitation has been acknowledged when discussing this result:

“However, this result must be interpreted with caution as it is possible that re-exposure to the devalued US in the magazine resulted in some form of short-term avoidance to the magazine that persisted throughout the subsequent test session when the devalued CS was presented.”

2) It remains the case that the findings of deficits in initial acquisition from posterior lOFC lesions makes it difficult to interpret a reversal learning deficit here.

We agree that the acquisition deficit does make the reversal deficit harder to interpret, but in general reversal learning procedures are quite complex and any deficit is hard to interpret without pursuing simpler follow up tasks to probe the nature of the deficit. Nonetheless, we have stressed that the acquisition deficit appears to affect both magazine approach and lever pressing behaviour whereas the reversal deficit selectively disrupts extinction of the magazine approach but not lever pressing. Therefore, we would argue that the two deficits are distinct.

3) There is very little consideration of the limitations of the pre-training lesion approach selected here. Encoding v. retrieval, compensatory mechanisms, etc. These limitations are important to understanding the present results and should be discussed. Also, in my opinion the discussion is too long. It reads more like a review paper than a discussion of the present results. I suggest making it much shorter.

This point is well taken, and we agree that there are benefits and limitations to pre-training lesion approaches that should be highlighted. This has been added to the Discussion section.

4) Figure 1D. I still think the devaluation effect is weak in the Sham group and worry these data might color the report for readers. Perhaps plotting the data over time might reveal a bigger effect earlier in the test.

In presenting the data as total session responding we aimed to keep the presentation of devaluation effect the same between experiments. To supplement this analysis we have included the data plotted and analysed over time as a figure supplement. This analysis reveals a main effect of Devaluation in the Lesion group, but a Devaluation x Time interaction in the Sham group. Again, the effect is not as strong in the Sham group, but hopefully this additional analysis and significant effect will help clarify the strength of the effect for the readers.